## ARTICLES

# Integrating de novo and inherited variants in 42,607 autism cases identifies mutations in new moderate-risk genes

Xueya Zhou[1,2,63], Pamela Feliciano[3,63], Chang Shu[1,2,63], Tianyun Wang[4,5,6,63], Irina Astrovskaya[3,63], Jacob B. Hall[3], Joseph U. Obiajulu[1,2], Jessica R. Wright[3], Shwetha C. Murali[4,7], Simon Xuming Xu[3], Leo Brueggeman[8], Taylor R. Thomas[8], Olena Marchenko[3], Christopher Fleisch[3], Sarah D. Barns[3], LeeAnne Green Snyder[3], Bing Han[3], Timothy S. Chang[9], Tychele N. Turner[10], William T. Harvey[4], Andrew Nishida[11], Brian J. O'Roak[11], Daniel H. Geschwind[9], The SPARK Consortium*, Jacob J. Michaelson[8], Natalia Volfovsky[3], Evan E. Eichler[4,7], Yufeng Shen[2,12,64] and Wendy K. Chung[1,3,13,64] ✉

**To capture the full spectrum of genetic risk for autism, we performed a two-stage analysis of rare de novo and inherited coding variants in 42,607 autism cases, including 35,130 new cases recruited online by SPARK. We identified 60 genes with exome-wide significance ($P < 2.5 \times 10^{-6}$), including five new risk genes (*NAV3*, *ITSN1*, *MARK2*, *SCAF1* and *HNRNPUL2*). The association of *NAV3* with autism risk is primarily driven by rare inherited loss-of-function (LoF) variants, with an estimated relative risk of 4, consistent with moderate effect. Autistic individuals with LoF variants in the four moderate-risk genes (*NAV3*, *ITSN1*, *SCAF1* and *HNRNPUL2*; $n = 95$) have less cognitive impairment than 129 autistic individuals with LoF variants in highly penetrant genes (*CHD8*, *SCN2A*, *ADNP*, *FOXP1* and *SHANK3*) (59% vs 88%, $P = 1.9 \times 10^{-6}$). Power calculations suggest that much larger numbers of autism cases are needed to identify additional moderate-risk genes.**

Many previous genetic studies in autism spectrum disorder (ASD), a neurodevelopmental condition characterized by social communication difficulties and repetitive behaviors[1], focused on de novo variants (DNVs) identified from parent–offspring trios[2–8]. Over 100 high-confidence ASD genes enriched with likely deleterious DNVs have been identified[8], most of which are also enriched for DNVs in other neurodevelopmental disorders (NDDs)[9–11]. Statistical modeling suggests that there are ~1,000 genes with DNVs in ASD[12,13]. However, despite the large effect size of individual pathogenic DNVs, all DNVs combined explain only ~2% of variance in liability for ASD[8,14]. ASD is highly heritable[14–16], and previous studies estimated that common variants explain up to half of the heritability[14], although only five genome-wide significant loci have been identified[17]. Rare LoF variants in genes intolerant of variation[9,18] are overtransmitted to probands compared with siblings without ASD[7,8,19–22]. However, identification of the individual risk genes enriched by such inherited variants has remained elusive. We have established the largest ASD cohort, Simons Foundation

Powering Autism Research for Knowledge (SPARK)[23], which currently includes over 100,000 people with ASD, to advance research on the genetic, behavioral and clinical features associated with ASD.

Rare LoF variants are enriched in developmental disorders including ASD[22,24], but may also result from sequencing and annotation artifacts[25] and present technical challenges in large sequencing studies. Methods to distinguish between high-confidence and low-confidence LoF variants[18,26,27] have been used to quantify gene-level LoF intolerance[18,26,28,29] and to refine the role of LoF DNVs in NDDs[20].

Here, we present an integrated analysis of de novo and inherited coding variants in over 42,607 ASD cases, including cases from previously published ASD cohorts and 35,130 new cases from SPARK. In our two-stage design, we first characterized the contribution of DNVs and rare inherited LoF variants to ASD risk. Results from the first stage informed the second stage meta-analysis of 404 genes. By combining evidence from DNVs, transmission disequilibrium tests (TDTs) and case-control comparisons, we identified 60 ASD risk

[1]Department of Pediatrics, Columbia University Medical Center, New York, NY, USA. [2]Department of Systems Biology, Columbia University Medical Center, New York, NY, USA. [3]Simons Foundation, New York, NY, USA. [4]Department of Genome Sciences, University of Washington School of Medicine, Seattle, WA, USA. [5]Department of Medical Genetics, Center for Medical Genetics, School of Basic Medical Sciences, Peking University Health Science Center, Beijing, China. [6]Neuroscience Research Institute, Department of Neurobiology, School of Basic Medical Sciences, Peking University Health Science Center; Key Laboratory for Neuroscience, Ministry of Education of China & National Health Commission of China, Beijing, China. [7]Howard Hughes Medical Institute, University of Washington, Seattle, WA, USA. [8]Department of Psychiatry, University of Iowa Carver College of Medicine, Iowa City, IA, USA. [9]Program in Neurogenetics, Department of Neurology, David Geffen School of Medicine, University of California, Los Angeles, Los Angeles, CA, USA. [10]Department of Genetics, Washington University, St. Louis, MO, USA. [11]Department of Molecular & Medical Genetics, Oregon Health & Science University, Portland, OR, USA. [12]Department of Biomedical Informatics, Columbia University Medical Center, New York, NY, USA. [13]Department of Medicine, Columbia University Medical Center, New York, NY, USA. [63]These authors contributed equally: Xueya Zhou, Pamela Feliciano, Chang Shu, Tianyun Wang, Irina Astrovskaya. [64]These authors jointly supervised this work: Yufeng Shen, Wendy K. Chung. *A list of authors and their affiliations appears at the end of the paper. ✉e-mail: wkc15@columbia.edu

genes with exome-wide significance, including five new genes not previously implicated in NDDs. Finally, we estimated the effect sizes of known and newly identified genes and conducted power calculations to inform the design of future studies.

## Results

**Overview of data and workflow.** We aggregated exome or whole genome sequencing (WGS) data of 35,130 new cases from SPARK and 7,665 cases from published ASD studies (ASC[3,8], MSSNG[6] and SSC[2,30]) (Supplementary Table 1) and performed a two-stage analysis (Fig. 1). In stage 1, we analyzed DNVs in 16,877 ASD trios and assessed transmission of rare LoF variants from 20,491 parents without ASD diagnoses or intellectual disability to offspring with ASD (including 9,504 trios and 2,966 single-parent-proband duos). For DNVs, we characterized the enrichment pattern in known and candidate risk genes, as well as mutation intolerance (probability of being LoF intolerant as defined by the Exome Aggregation Consortium (ExAC pLI[18], and Genome Aggregation Database (gnomAD) metrics[26]), and performed gene-based burden tests of LoF and missense DNVs by DeNovoWEST[11]. For rare inherited LoFs, we estimated the overtransmission from unaffected parents to ASD offspring in all genes and gene sets predefined by functional genomic data or results from DNV analysis. Based on DNV enrichment and overtransmission patterns in gene sets, we selected 404 genes for meta-analysis in stage 2 using 22,764 new cases with exome or WGS data. In stage 2, we applied DeNovoWEST on DNVs, conducted TDTs on inherited LoFs in trios or duos, performed burden tests on rare LoFs in unrelated cases compared with population controls (104,068 subjects from non-neuro gnomAD exomes and 132,345 TOPMed subjects) and combined the P values to estimate a final P value for each of the 404 genes. Finally, we performed a mega-analysis of rare LoFs in all cases and controls to estimate the effect sizes of known or new candidate ASD genes to inform future studies.

**Known ASD or NDD risk genes explain most de novo burden.** In the first stage, we combined data from four large-scale ASD cohorts, including 16,877 unique ASD trios and 5,764 unaffected trios (Supplementary Table 1). The cohorts show similar exome-wide burden of DNVs in simplex families. The burden of LoF DNVs in cases with an ASD family history is significantly lower than those without ($P = 1.1 \times 10^{-4}$ by Poisson test), whereas the burden of predicted de novo damaging missense (D-mis, defined by rare exome variant ensemble learner (REVEL) score[31] $\geq 0.5$) and synonymous variants are similar (Extended Data Fig. 1). Compared with unaffected offspring, the excess of damaging DNVs (de novo LoF and D-mis variants) in individuals with ASD is concentrated in LoF-intolerant genes, defined as genes with an ExAC pLI $\geq 0.5$ (ref. [18]). Using LoF observed/expected upper-bound fraction (LOEUF), a recently developed gene constraint metric[26], the burden of damaging DNVs is highest among genes ranked in the top 20% of LOEUF scores (Fig. 2a). Overall, the population attributable risk (PAR) from damaging DNVs is about 10%. We assembled 618 previously established dominant ('known') ASD or NDD risk genes (Supplementary Table 2). These genes explained about two-thirds of the PAR from damaging DNVs. Excluding these genes, the fold enrichment of damaging DNVs was greatly attenuated (Fig. 2a).

To assess the evidence of DNVs in individual genes, we applied DeNovoWEST[11], which integrates DNV enrichment with clustering of missense variants in each gene. The initial DeNovoWEST scan of DNVs in 16,877 ASD trios identified 159 genes with $P < 0.001$ (Supplementary Table 3).

**Rare inherited LoFs are mostly in unknown ASD risk genes.** To analyze the contribution of rare inherited LoF variants to ASD risk, we evaluated transmission disequilibrium in ultra-rare (allele frequency $< 1 \times 10^{-5}$) high-confidence (by the loss-of-function

transcript effect estimator (LOFTEE)[26] package and proportion expression across transcripts (pExt)[27]; see Methods and Supplementary Note) LoF variants from parents without ASD diagnoses or intellectual disability to affected offspring with ASD in 9,504 trios and 2,966 duos from the first stage (Supplementary Table 4). For a given set of genes, we quantified transmission disequilibrium using the number of overtransmitted (excess in transmission over nontransmission) LoF variants per trio; parent–offspring duos were considered half-trios. Among autosomal genes, the overall transmission disequilibrium signal of ultra-rare LoF variants is enriched in LoF-intolerant genes (ExAC pLI $\geq 0.5$) and in genes within the top 20% of LOEUF scores (Fig. 2b), similar to the burden of damaging DNVs. We observed both overtransmission to affected and undertransmission to unaffected offspring, especially in genes within the top 10% of LOEUF scores. However, known ASD or NDD genes explain only ~20% of overtransmission of LoF variants to affected offspring (Fig. 2b). On the X chromosome, we only considered transmission from mothers without ASD to 9,883 affected sons and 2,571 affected daughters (Supplementary Table 4). Rare LoF variants in mothers without ASD show significant overtransmission to affected sons but not affected daughters and remain significant after removing known ASD or NDD genes (Supplementary Fig. 1). Together, these results suggest that most genes conferring inherited ASD risk are yet to be identified. Autosomal rare D-mis variants also show evidence of transmission disequilibrium to affected offspring, although the signal is much weaker (Supplementary Fig. 2).

To characterize the properties of genes contributing to ASD risk through rare inherited variants, we defined 25 gene sets from five categories representing both functional and genetic evidence relevant to ASD (Supplementary Table 5 and Supplementary Fig. 3). We limited the genes to 5,754 autosomal constrained genes (ExAC pLI $\geq 0.5$ or top 20% of LOEUF scores) and performed TDT (Supplementary Table 6). For each gene set, we tested if ultra-rare high-confidence rare LoF variants show a higher transmission to ASD offspring than the remaining genes in the overall constrained gene set. As a comparison with DNVs, we also tested if the same set of genes are more frequently disrupted by damaging DNVs than the rest of the genes in ASD trios using DNENRICH[32].

Using functional gene sets derived from the neuronal transcriptome, proteome or regulome, we confirmed significant enrichment in damaging DNVs ($P < 0.005$ by simulation) in the gene sets that were previously suggested to be enriched for ASD risk genes including expression module M2/3[33], RBFOX1/3 targets[34], FMRP targets[35] and CHD8 targets[36]. However, this enrichment can be largely explained by known ASD or NDD genes (Extended Data Fig. 2). For ultra-rare inherited LoF variants, we found that the proportion of transmission to ASD individuals in most functional gene sets is close to all other genes; only RBFOX targets show a weak enrichment but can be largely explained by known genes (Fig. 3). We also applied two machine learning methods to prioritize ASD risk genes: forecASD[37] and A-risk[38]. Although enrichment of DNVs in predicted genes is mainly explained by known genes, genes prioritized by A-risk are significantly enriched with inherited LoFs that are not explained by known genes. Using A-risk $\geq 0.4$, 30% of constrained genes ($n = 1,464$) were prioritized and explain 64% of the overtransmission of LoF variants to ASD offspring ($P = 2.6 \times 10^{-5}$ by $\chi^2$ test). This enrichment is higher than genes prioritized by the LOEUF score; 33% of genes ($n = 1,777$) in the top decile of LOEUF account for 55% of the overtransmission ($P = 3.5 \times 10^{-4}$ by $\chi^2$ test) (Fig. 3).

We also considered gene sets that have evidence of genetic association with DNVs. Genes nominally enriched by DNVs ($P < 0.01$ by DeNovoWEST; $n = 300$) in ASD from the current study have a significantly higher overtransmission rate than other constrained genes (odds ratio $= 1.39$, $P = 3.0 \times 10^{-5}$ by $\chi^2$ test) (Fig. 3), although these genes account for only 21% of the overtransmission. Genes

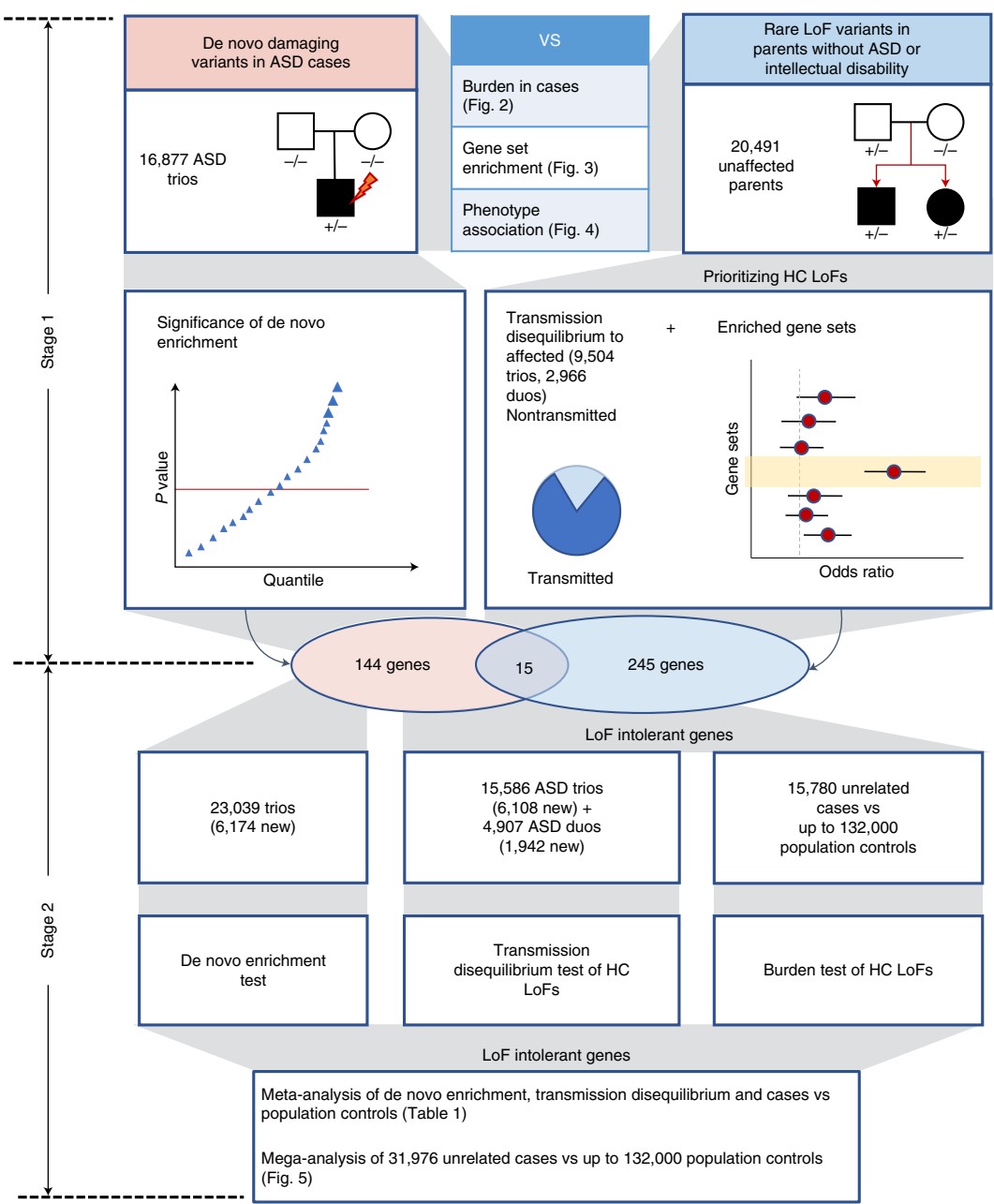

**Fig. 1 | Analysis workflow.** In the discovery stage, we identified DNVs in 16,877 ASD trios and rare LoF variants in 20,491 parents without ASD diagnoses and intellectual disability. We compared properties of de novo and rare variants to identify rare LoFs that contribute to genetic risk in individuals with ASD. We also evaluated their associations with cognitive impairment and enriched gene sets. We performed an initial exome-wide scan of genes enriched by DNVs or showing transmission disequilibrium of rare LoFs to affected offspring and selected a total of 404 genes for further replication, including 159 de novo enriched genes and 260 prioritized transmission disequilibrium genes from enriched gene sets (15 genes were in both). In the meta-analysis stage, we first evaluated evidence from de novo enrichment and transmission disequilibrium of rare inherited LoFs in an expanded set of family-based samples including over 6,000 additional ASD trios and around 2,000 additional duos. The DNVs in ASD were combined with those from an additional 31,565 NDD trios to refine the filters of high-confidence LoF variants in de novo LoF enriched genes. We also constructed an independent dataset of LoF variants of unknown inheritance from 15,780 cases that were not used in de novo or transmission analysis. We compared LoF rates in cases with two population-based sets of controls ($n = {\sim}104{,}000$ and ${\sim}132{,}000$, respectively). For 367 LoF-intolerant genes on autosomes, the final gene-level evidence was obtained by meta-analyzing $P$ values of de novo enrichment, transmission disequilibrium of high-confidence rare inherited LoFs, and comparison of high-confidence LoFs from cases and controls not used in the de novo or transmission analysis. We also performed a mega-analysis that analyzed high-confidence LoFs identified in all 31,976 unrelated ASD cases and compared their rates with population-based controls. HC, high-confidence.

nominally enriched by DNVs in other NDDs[11] are also significantly enriched by DNVs in ASD and weakly enriched by inherited LoFs in ASD; however, both can be largely explained by known genes

(Fig. 3). This suggests that a subset of ASD genes increase risk by both DNVs and inherited variants, and new genes can be identified by integrating evidence from DNV enrichment and TDT.

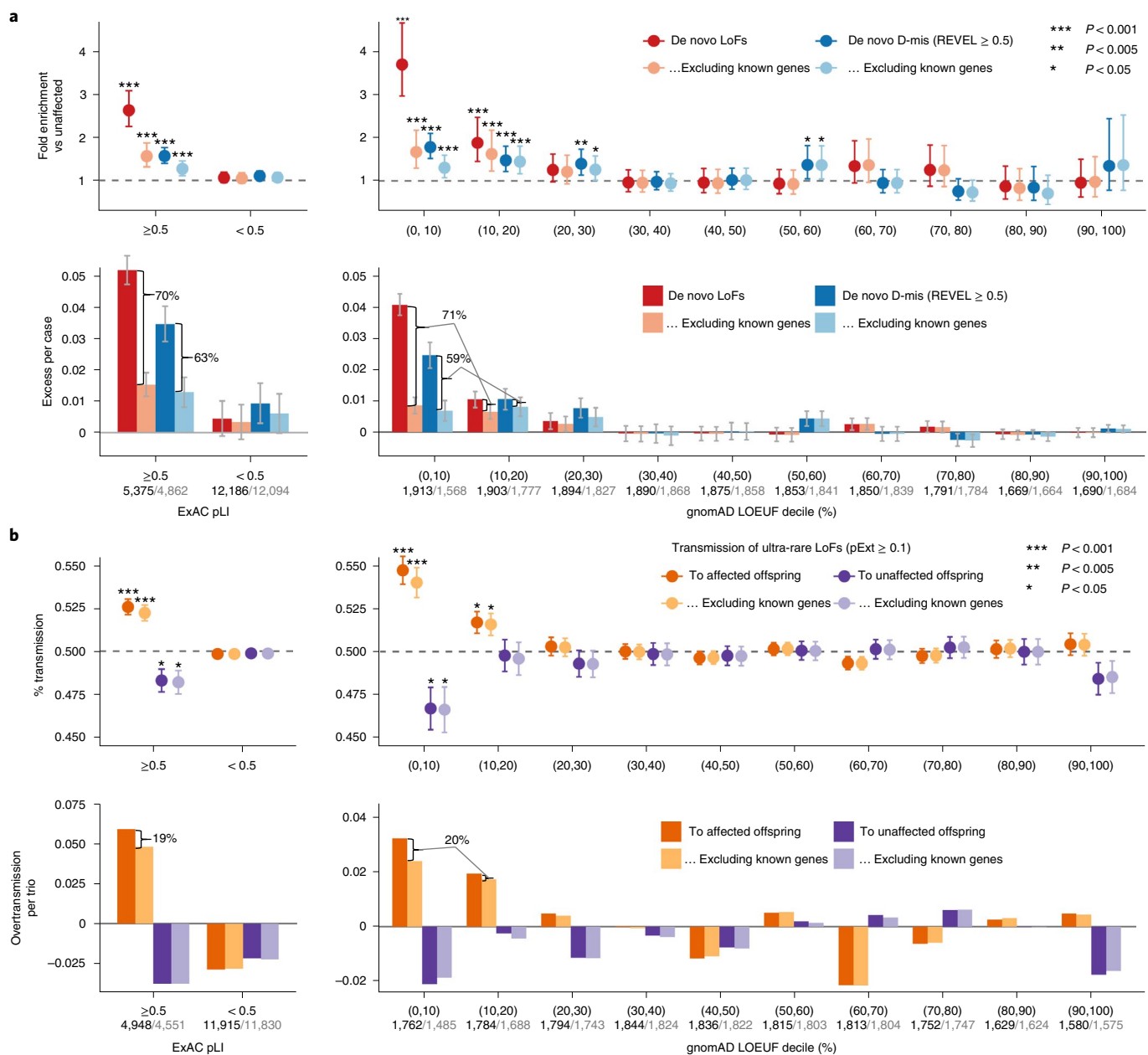

**Fig. 2 | Comparison of burden between de novo damaging variants and rare inherited LoFs in ASD. a**, The burden of DNVs was evaluated by the rate ratio and rate difference between 16,877 ASD and 5,764 unaffected trios. The exome-wide burden of de novo LoF and D-mis (REVEL ≥ 0.5) variants are concentrated in constrained genes (ExAC pLI ≥ 0.5) and in genes with the highest levels of LoF intolerance in the population (defined by the top two deciles of gnomAD LOEUF scores). Burden analysis was repeated after removing known ASD or NDD genes. The number of genes before and after removing known genes in each constraint bin is shown below the axis label. Data are presented as mean values and 95% confidence intervals. Among constrained genes (ExAC pLI ≥ 0.5 or the top 20% of gnomAD LOEUF scores), close to two-thirds of case-control rate differences of de novo LoF and D-mis variants can be explained by known genes. Exact P values by Poisson test are listed in Supplementary Table 19. **b**, The burden of inherited LoFs was evaluated by looking at the proportion of rare LoFs in 20,491 parents without ASD diagnoses or intellectual disability that are transmitted to affected offspring in 9,504 trios and 2,966 duos and show evidence of overtransmission of LoFs per ASD trio. As a comparison, we also show the transmission disequilibrium pattern to unaffected offspring in 5,110 trios and 129 duos. Data are presented as mean values ± standard errors as error bars. Two-sided binomial test was used to compute the P values for overtransmission or undertransmission. Using ultra-rare LoFs with pExt ≥ 0.1, exome-wide signals of transmission disequilibrium of rare inherited LoF variants also concentrate in constrained genes (ExAC pLI ≥ 0.5) and in genes within the top two deciles of gnomAD LOEUF scores. Analysis was restricted to autosomal genes and repeated after removing known ASD or NDD genes (number of genes in each constrained bin before and after removing known genes is shown below the axis label). Among all constrained genes, only one-fifth of overtransmission of LoFs to ASD trios can be explained by known ASD or NDD genes. Exact P values by binominal test are listed in Supplementary Table 19.

**DNVs and some rare inherited LoFs are associated with intellectual disability.** To evaluate the association of genotypes with phenotype in ASD, we used self-reported cognitive impairment in SPARK,

a Vineland score of <70 in the SSC or the presence of intellectual disability in ASC. Damaging DNVs in genes ranked within the top 10% of LOEUF scores show a higher burden ($P = 1.1 \times 10^{-24}$ by $\chi^2$

test) in ASD cases with evidence of cognitive impairment than in other cases, consistent with previous results[2,8] (Fig. 4a). Once known ASD or NDD genes were excluded, the residual burden of damaging DNVs in genes within the top 10% of LOEUF scores is greatly reduced and not significantly associated with cognitive phenotype in ASD (Fig. 4a). Overtransmission of rare LoFs in genes within the top 10% of LOEUF genes to ASD cases with cognitive impairment is about 2.7 times higher than to cases without cognitive impairment ($P = 4.6 \times 10^{-3}$ by $\chi^2$ test) and is still 2 times higher ($P = 0.04$ by $\chi^2$ test) once known ASD or NDD genes were excluded (Fig. 4b). However, rare LoFs in genes prioritized by A-risk are not associated with cognitive impairment (Supplementary Fig. 4). Taken together, these results suggest that rare variants in the top 10% of LOEUF genes—most of which are already known to be ASD or NDD risk genes—are associated with cognitive impairment. However, a subset of rare inherited variants, particularly those prioritized by A-risk, are not associated with cognitive impairment.

**Meta-analysis identifies five new risk genes.** Based on results from the first stage, we identified 260 genes with evidence of TDT (TDT statistic[39] $\geq 1$) and in gene sets enriched with rare inherited LoFs (top 10% LOEUF or within top 20% LOEUF and A-risk $\geq 0.4$) (Supplementary Table 6) and 159 genes with $P < 0.001$ from the DeNovoWEST analysis of DNVs (with 15 genes by both) (Supplementary Table 3). We performed a meta-analysis on the 367 autosomal genes with all data from stage 1 and stage 2, which includes 6,174 new ASD trios, 1,942 new duos, 15,780 unrelated cases (see Methods) and 236,000 population controls.

We used Fisher's method[40] to combine three $P$ values that estimate independent evidence of DNVs, TDT and case-control comparison: (1) DeNovoWEST with DNVs from both stage 1 and stage 2 ($n = 23,039$ trios, Supplementary Tables 1 and 7) using the parameters estimated in stage 1, (2) TDT with rare LoF variants in parents without ASD diagnoses or intellectual disability with affected offspring in 15,586 trios and 4,907 duos (Supplementary Table 4) and (3) unrelated cases (Supplementary Table 8) compared with population controls using a binomial test. We used two sets of controls: gnomAD exome v2.1.1 non-neuro subset (only samples from individuals who were not ascertained for having a neurological condition in a neurological case-control study, $n = 104,068$) and TOPMed WGS (freeze 8, $n = 132,345$). We performed a case-control burden test using the two sets separately and input the larger $P$ value for the meta-analysis. This approach avoids sample overlap and helps ensure that significant genes are not dependent on the choice of population reference. Although population reference data were processed by different pipelines, the cumulative allele frequencies (CAFs) of high-confidence LoF variants (see Methods) are similar between internal pseudocontrols (see Methods) and the two population references after applying the same LoF filters (Supplementary Fig. 5). Previous population genetic simulations predict that for genes under moderate to strong selection (selection coefficient > 0.001), deleterious variants are expected to arise within 1,000 generations and population demographic histories do not confound the CAFs of deleterious alleles in these genes[41].

For 367 selected autosomal genes, the point estimates of selection coefficient under the mutation–selection balance model[42] are all greater than 0.01 (Supplementary Fig. 6). Most high-confidence

LoF variants in these genes are ultra-rare (Supplementary Fig. 7), and the CAFs of high-confidence LoF variants in European and non-European population samples are highly correlated (Supplementary Fig. 8). Therefore, we included population samples across all ancestries as controls. The ultra-rare synonymous variant burden is similar between cases and controls across the selected genes (Extended Data Fig. 3). To make use of all genetic data collected, we also included rare variants of unknown inheritance from ASD cases that were analyzed in the first stage. These variants come from cases that are part of unaffected parent–ASD duos; such variants were either inherited from the parent not participating in the study or DNVs. Therefore, these variants are independent of TDT, even though the same cases were included in TDT.

We identified 60 genes with exome-wide significance ($P < 2.5 \times 10^{-6}$), and 72 genes reached study-wide significance accounting for all 5,754 constrained genes ($P < 8.7 \times 10^{-6}$, Supplementary Table 9). Figure 5 summarizes the distribution of LoF variants (with different modes of inheritance) in genes that reached study-wide significance by DNV enrichment (Fig. 5a) and other significant genes by meta-analysis (Fig. 5b and Supplementary Fig. 9). Genes that are significant only in meta-analysis tend to harbor more inherited LoF variants than DNVs, consistent with their lower penetrance for ASD or NDD.

Although most significant genes were previously known, we identified five new genes that have exome-wide significance regardless of the choice of population reference: *NAV3*, *MARK2*, *ITSN1*, *SCAF1* and *HNRNPUL2* (Table 1). The combined $P$ values based on ancestry-specific case-control analyses are similar to the overall case-control analysis for these five genes (Supplementary Table 10). As expected, most supporting variants are ultra-rare, and results are robust to the allele frequency filter. These five new genes together explain 0.27% of the PAR ratio (Supplementary Table 11). *NAV3* has a similar PAR to that of *CHD8* and *SCN2A* (~0.095%). *ITSN1* is similar to *PTEN* (~0.065%).

The association of *NAV3* with ASD risk is primarily driven by rare inherited variants (Table 1). *NAV3* has a high A-risk score, suggesting that the expression pattern of *NAV3* is highly similar to known ASD genes (Supplementary Data 1)[7,43]. *NAV3* has high expression in the inner cortical plate of the developing cortex[33], and in pyramidal neurons (hippocampus CA1 and somatosensory cortex) and cortical interneurons[44,45] (Supplementary Fig. 10). The association of *MARK2* with ASD risk is primarily driven by DNVs and is also associated with other NDDs[11] ($P = 2.7 \times 10^{-5}$ by DeNovoWEST) including Tourette syndrome[46] and epilepsy[47]. We find that three out of eight autistic offspring with variants in *MARK2* report epilepsy, two out of eight report Tourette syndrome and seven out of eight have evidence of cognitive impairment (Supplementary Table 12).

The remaining three novel genes have support from both DNVs and rare LoFs. *ITSN1* and *SCAF1* show nominal significance of DNV enrichment in 31,058 NDD trios[11] ($P < 0.05$ by DeNovoWEST). *SCAF1* was among the top 50 genes from a gene-based burden test in a recent schizophrenia case-control study ($P = 0.0027$ by burden test)[48]. Both *ITSN1* and *NAV3* have moderate effect sizes (point estimate of relative risk 3~6; Supplementary Table 11). *ITSN1* has been highlighted in our previous study with evidence of enriched inherited LoFs[7]. We also assessed deletions in these new genes. For both *ITSN1* and *NAV3*, we identified four partial or whole gene deletions

**Fig. 3 | Enrichment of rare LoF variants in ASD cases across gene sets.** Gene sets were defined and grouped by transcriptome proteome, neuronal regulome, ASD gene prediction scores, genetic evidence from neuropsychiatric diseases, and gene-level constraint. Analyses were repeated after removing known ASD or NDD genes. (Number of genes in each set before and after removing known genes are shown in parentheses below gene set.) Dots represent fold enrichment of DNVs or odds ratios for overtransmission of LoF variants in each set. Horizontal bars are presented as mean values with 95% confidence interval as error bars. For each gene set, we show the percentage of overtransmission of rare LoFs to cases. Enrichment of rare inherited LoFs was evaluated by the share of overtransmission events (the transmission and nontransmission of ultra-rare LoFs with pExt $\geq 0.1$) in the selected gene set vs those in all other constrained genes using a two-by-two table. $P$ values were determined using the $\chi^2$ test. Exact $P$ values are listed in Supplementary Table 19.

Transmission of rare, inherited LoFs

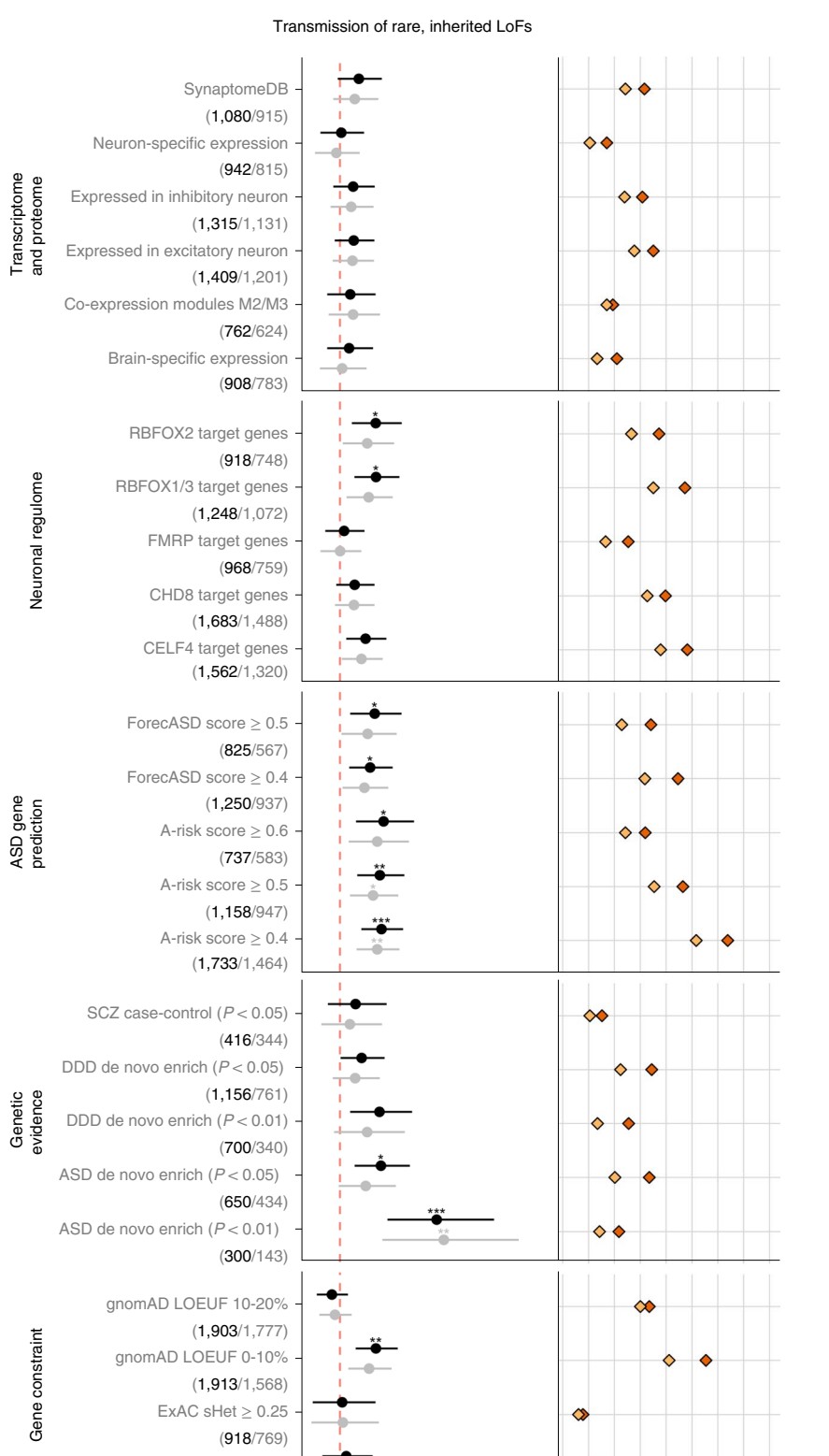

*** $P < 5 \times 10^{-5}$ — ● All constrained genes ◆ Ultra-rare LoFs (pExt ≥ 0.1) in unaffected parents

** $P < 5 \times 10^{-4}$ — ● … Excluding known genes ◇ … Excluding known genes

* $P < 0.005$

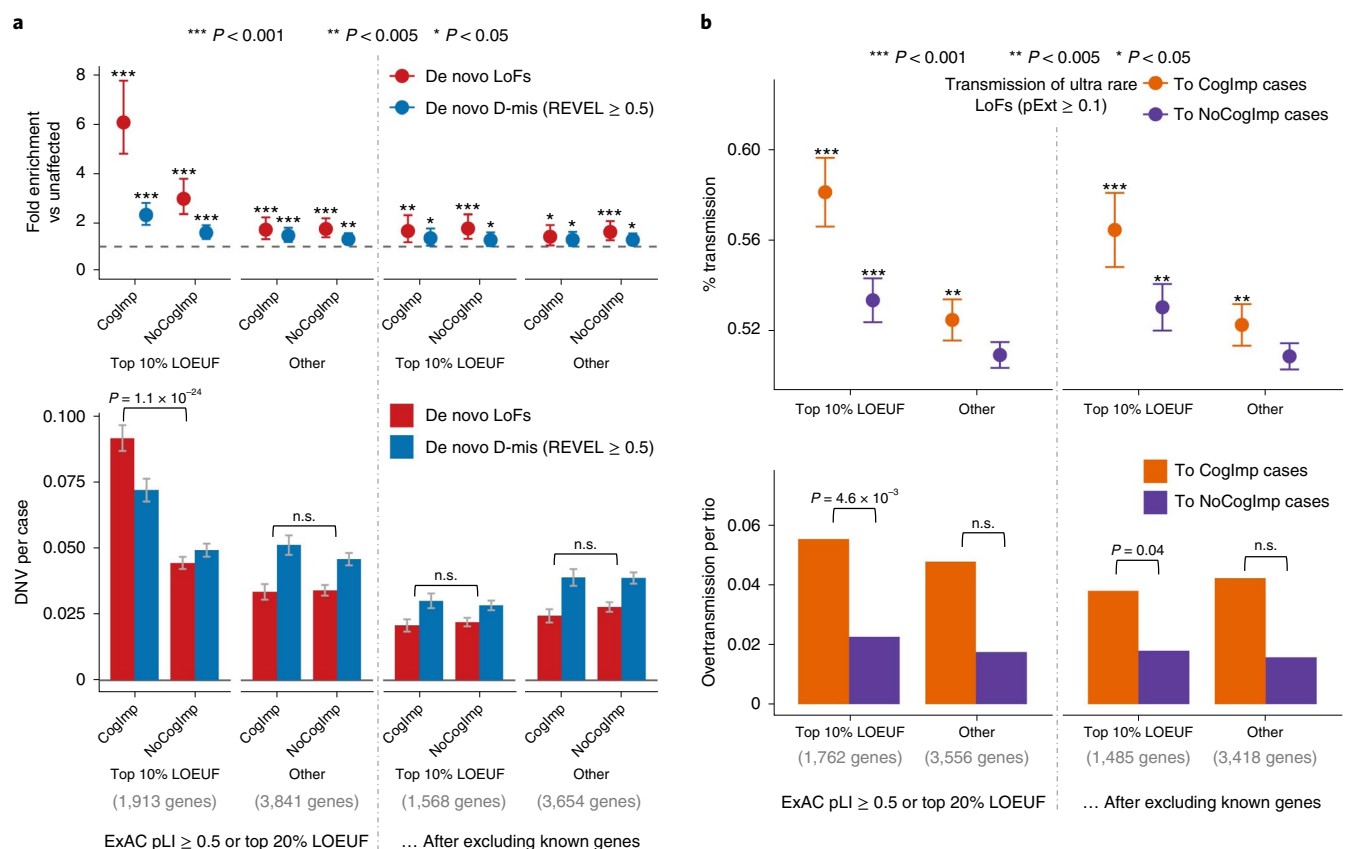

**Fig. 4 | Association of rare inherited LoFs with cognitive impairment in ASD cases.** Ultra-rare inherited LoFs with pExt ≥ 0.1 in genes with the top 10% of gnomAD LOEUF scores also show a higher proportion of transmission and a higher overtransmission rate to ASD offspring with cognitive impairment (CogImp) than those without (NoCogImp). Rare LoFs in other constrained genes are not significantly associated with phenotypic severity. The increased burden of inherited LoFs in cases with cognitive impairment remains significant after removing known ASD or NDD genes. Data are presented as mean values ± standard errors as error bars. Poisson test was used to compute the P values to assess the fold enrichment, and binominal test was used for overtransmission. Exact P values are listed in Supplementary Table 19.

in 33,083 parents without ASD diagnoses or intellectual disability that also show transmission disequilibrium to affected offspring (Extended Data Fig. 4).

Although both de novo and rare inherited LoFs in the most constrained genes are strongly associated with intellectual disability in ASD (Fig. 4), the association of such variants in individual genes is heterogenous, as suggested by the lack of association of rare inherited variants in genes with high A-risk scores (Extended Data Fig. 2). We calculated the burden of cognitive impairment (see Methods) in 87 ASD individuals with high-confidence LoF variants in the four novel moderate-risk genes and compared it with that in 129 individuals with high-confidence LoF in the well-established ASD risk genes *CHD8, SCN2A, SHANK3, ADNP* and *FOXP1*, as well as 8,731 individuals with ASD (Supplementary Fig. 11). Although most individuals with variants in well-established ASD risk genes have some evidence of cognitive impairment (88%), individuals with LoF variants in the moderate-risk genes had significantly lower burden (56%, $P = 4.5 \times 10^{-7}$). Individuals with high-confidence LOFs in the moderate-risk genes did not have a significantly different burden of cognitive impairment than 8,731 individuals with ASD in SPARK (56% vs 50%, P = n.s.). Individuals with LoF variants in the moderate-risk genes also had a similar male:female (4:1) ratio compared with the larger cohort, whereas individuals with variants in the well-established ASD risk genes showed significantly less male bias (1.6:1, P = 0.009) (Supplementary Fig. 11), as previously reported[2]. We also predicted full-scale intelligence quotient (IQ) on all participants based on parent-reported data using a machine

learning method[49]. Heterozygotes for rare LoFs in three (*NAV3, SCAF1* and *HNRNPUL2*) of the four new genes with substantial contribution from rare inherited variants have similar IQ distribution as the overall SPARK cohort (Fig. 6a), which is substantially higher than heterozygotes with rare LoFs in well-established, highly penetrant genes that contribute to ASD primarily through DNVs ('DN genes') such as *CHD8, SHANK3* and *SCN2A*. In fact, both novel and established genes with significant contribution from rare inherited LoFs are less associated with intellectual disability than NDD genes (Fig. 6b). Across these genes, there is a significant negative correlation ($r = 0.78$, $P = 0.001$) of estimated relative risk of rare LoFs with average predicted IQ of the individuals with these variants (Fig. 6c).

Most known ASD or NDD genes that are enriched by LoF DNVs harbor more de novo than LoF inherited variants in ~16,000 unrelated ASD trios (Fig. 5a and Supplementary Fig. 16), consistent with their high penetrance for ASD or NDD phenotypes and strong negative selection. Using population exome or WGS data, we calculated a point estimate of selection coefficient ($\hat{s}$)[50] of LoFs in each gene (Supplementary Table 11) and found that the fraction of de novo LoFs in ASD genes is higher in genes with large $\hat{s}$, and smaller in genes with small $\hat{s}$ (Supplementary Fig. 5b), consistent with population genetic theory[51]. We also estimated average effect size of rare LoFs in ASD genes by comparing CAF in 31,976 unrelated cases and population exome or WGS data. As expected, known and newly significant ASD genes with higher risk for ASD are under stronger selection (larger $\hat{s}$) (Supplementary Fig. 13).

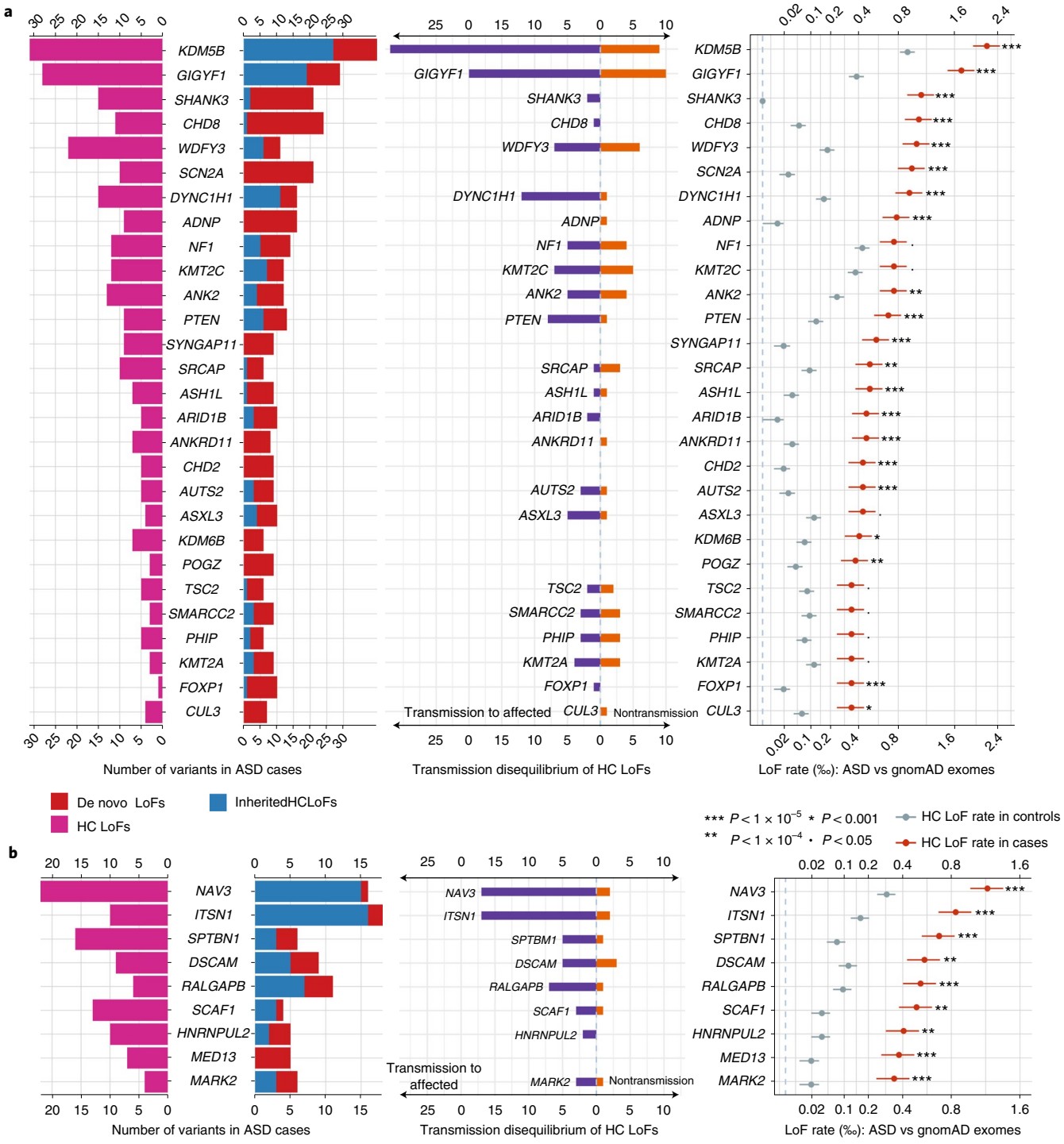

**Fig. 5 | Distribution of de novo and inherited LoF variants in known and novel ASD genes in cases and population controls.** From left to right: pyramid plots summarizing the number of de novo LoFs in 15,857 ASD trios, inherited high-confidence LoFs in 18,720 unrelated offspring included in transmission analysis, and high-confidence LoFs in 15,780 unrelated cases; bar plot of transmission vs nontransmission for rare high-confidence LoFs identified in parents without ASD diagnoses or intellectual disability; three plots comparing the high-confidence LoF rate in 31,976 unrelated ASD cases with gnomAD exomes (non-neuro subset, 104,068 individuals). Horizontal bars are presented as mean values ± standard errors as error bars. **a**, Twenty-eight known ASD or NDD genes that have LOEUF scores in the top 30% of gnomAD, have a $P$ value for enrichment among all DNVs ($P < 9 \times 10^{-6}$) in 23,039 ASD trios, and have more than 10 LoFs. **b**, Nine additional ASD risk genes that achieved a $P$ value of $<9 \times 10^{-6}$ in stage 2 of this analysis. The majority of genes in **b** harbor more inherited LoFs than DNVs. All five novel genes (Table 1) are shown in **b**. Note that the $x$ axes of LoF rates are in the squared root scale. Poisson test was used to compute the $P$ values. Exact $P$ values are listed in Supplementary Table 6.

**Functional similarity of new genes and known ASD genes.** To better appreciate the probable functional implications of the new exome-wide-significant genes that confer inherited risk for ASD,

we integrated mechanistic (STRING[52]) and phenotypic (Human Phenotype Ontology (HPO)[53]) data into a single embedding space (six dimensions, one for each archetype coefficient) using a com-

**Table 1 | Statistical evidence for the five novel exome-wide-significant ASD risk genes identified in this study**

| Gene | Prioritization | Enrichment of de novo damaging variants | | | | | Transmission disequilibrium of high-confidence LoFs | | | Case-control comparison of high-confidence LoF rate | | | $P_{Meta}$ |
|---|---|---|---|---|---|---|---|---|---|---|---|---|---|
| | | dnLoF | $\mu_{LoF}$ | dnD-mis | $\mu_{D\text{-mis}}$ | $P_{DNV}$ | Count | Trans: nontrans to affected | $P_{TDT}$ | Number (rate) of LoFs in cases | Rate of LoFs in controls: gnomAD exome, TOPMed | $P_{CC}$ | |
| NAV3 | TDT | 1 | $1.1\times10^{-5}$ | 1 | $1.1\times10^{-5}$ | 0.23 | 17 | 17:2 | $3.6\times10^{-4}$ | 22 ($1.4\times10^{-3}$) | $3\times10^{-4}$, $2.6\times10^{-4}$ | $4.4\times10^{-7}$, $2.1\times10^{-8}$ | $1.2\times10^{-8}$ |
| MARK2 | De novo | 5 | $4.4\times10^{-6}$ | 3 | $4.8\times10^{-6}$ | $8.9\times10^{-9}$ | 3 | 3:1 | 0.31 | 4 ($2.5\times10^{-4}$) | $2\times10^{-5}$, $6\times10^{-5}$ | $4.5\times10^{-3}$, 0.03 | $2.3\times10^{-8}$ |
| SCAF1 | TDT | 2 | $4.8\times10^{-6}$ | 0 | $1.7\times10^{-7}$ | $1.3\times10^{-3}$ | 4 | 3:1 | 0.31 | 11 ($7.0\times10^{-4}$) | $3\times10^{-5}$, $7\times10^{-5}$ | $2.1\times10^{-6}$, $1.4\times10^{-6}$ | $2.1\times10^{-7}$ |
| ITSN1 | TDT | 3 | $1.2\times10^{-5}$ | 2 | $1.3\times10^{-5}$ | $2.6\times10^{-3}$ | 18 | 17:2 | $3.6\times10^{-4}$ | 10 ($6.3\times10^{-4}$) | $1.6\times10^{-4}$, $2\times10^{-4}$ | $2\times10^{-3}$, $4\times10^{-3}$ | $4.3\times10^{-7}$ |
| HNRNPUL2 | De novo | 3 | $5.8\times10^{-6}$ | 0 | $3.8\times10^{-6}$ | $1.8\times10^{-3}$ | 2 | 2:0 | 0.25 | 10 ($6.3\times10^{-4}$) | $4\times10^{-5}$, $5\times10^{-5}$ | $2.6\times10^{-6}$, $8.2\times10^{-7}$ | $2.7\times10^{-7}$ |

Control high-confidence LoF rates are estimated from two population-based reference panels: gnomAD exome (v2.1.1, non-neuro subset, 104,068 individuals), and TOPMed (freeze 8, 132,345 individuals). Meta-analysis is done by combining $P$ values from de novo (DNV), transmission disequilibrium test (TDT) and case-control (CC) analysis using Fisher's method ($P_{Meta}$). For case-control, we conservatively took the largest $P$ value for meta-analysis. dnLoF, count of de novo loss-of-function variants; $\mu_{LoF}$, estimated background mutation rate of loss-of-function variants of the gene per chromosome; dnD-mis, count of de novo damaging missense variants, defined by REVEL score[31] ≥ 0.5); $\mu_{D\text{-mis}}$, estimated background mutation rate of rate damaging missense variants of the gene per chromosome; $P_{DNV}$, one-sided $P$ value for enrichment of all DNVs in 23,053 ASD trios; Trans: nontrans to affected, the ratio of transmitted vs. non-transmitted variants to affected offspring; $P_{TDT}$, one-sided $P$ value of overtransmission of high-confidence LoFs to affected offspring in 28,556 trios and 4,526 duos; $P_{CC}$, one-side $P$ value for increased high-confidence LoF rate in 15,811 unrelated cases compared with population controls (showing two $P$ values from comparison with gnomAD exome and TOPMed data, respectively).

bination of canonical correlation analysis and archetypal analysis (see Methods). This embedding space serves as an interpretive framework for putative ASD risk genes ($n = 1,776$). Six functional or phenotypic archetypes were identified (Fig. 7 and Supplementary Tables 13–15) that represent pathways that are well understood to play a role in ASD: neurotransmission (archetype 1 or A1), chromatin modification (archetype 2 or A2), RNA processing (archetype 3 or A3), vesicle-mediated transport (archetype 4 or A4), MAPK signaling and migration (archetype 5 or A5), and cytoskeleton and mitosis (archetype 6 or A6), also enriched for intermediate filaments. These archetypes organize risk genes in a way that jointly maximizes their association with mechanisms (STRING clusters) and phenotypes (HPO terms). For instance, A1 genes (neurotransmission) are enriched for the STRING cluster CL:8435 (ion channel and neuronal system) and are also associated with seizure and epileptic phenotypes. A2 genes (chromatin modifiers) are enriched for nuclear factors and genes linked to growth and morphological phenotypes (Supplementary Table 14). We call genes that strongly map to an archetype (that is, >2× the next highest-ranking archetype) 'archetypal'; if this criterion is not met, we call the genes 'mixed'. Archetypal genes are generally less functionally ambiguous than mixed genes. Of the five novel inherited risk genes, two are archetypal (suggesting function within known risk mechanisms): *NAV3* (A6, cytoskeleton and mitosis) and *ITSN1* (A4, vesicle-mediated transport). *SCAF1*, *MARK2* and *HNRNPUL2* are mixtures of the identified archetypes, largely A4 and A5. That these new genes did not resolve clearly into archetypes (that were defined by known and suspected autism risk genes) suggests that they may operate in potentially novel mechanisms. To elucidate these possibilities, we constructed an ad hoc archetype, defined by the centroid between *SCAF1*, *MARK2* and *HNRNPUL2* (see Fig. 7c). Cell–cell junction (CL:6549) was the STRING cluster most associated with this centroid ($P = 4.12 \times 10^{-14}$ by the Kolmogorov–Smirnov test; Fig. 7d), which fits with its location between A4 (vesicle-mediated transport) and A5 (MAPK signaling and migration).

**Power analysis.** The power of identifying risk genes with rare inherited variants or DNVs monotonically increases with increasing effect size or expected CAF under the null. New ASD genes to be discovered are likely to have smaller effect size than known ASD genes, as suggested by our results. Additionally, known ASD genes are biased toward longer genes, which have a higher background mutation rate of damaging variants ('long genes') (Extended Data Fig. 5). Even though longer genes are more likely to be expressed in the brain and be relevant to ASD or NDD[54], among most constrained genes, long genes (LoF mutation rate[55,56] above 80% quantile) and short genes (below 80%) have similar enrichment of damaging DNVs and rare inherited LoFs (Supplementary Fig. 14). Notably, for short genes, known genes have virtually no contribution to overtransmitted high-confidence LoFs to affected offspring (Supplementary Fig. 14b), suggesting that many short ASD risk genes remain to be identified.

We used a published framework[41] to analyze power based on case-control association of rare variants. For rare variants in genes under strong selection, CAF is largely determined by mutation rate and selection coefficient[41]. We therefore modeled power of discovering risk genes as a function of relative risk and selection coefficient. With about 5,500 constrained genes, the power of the current study was calculated for 31,976 unrelated cases and an experiment-wise error rate of $9 \times 10^{-6}$ (Extended Data Fig. 6).

We inversed the power calculation to determine the required sample size to achieve 90% power under the same assumptions (Extended Data Fig. 7). For genes at median LoF mutation rate across all genes, we estimated that it requires about 96,000 cases (three times the current sample size) to identify genes with similar effect size as *NAV3* (relative risk = 4.5) and *ITSN1* (relative risk = 5),

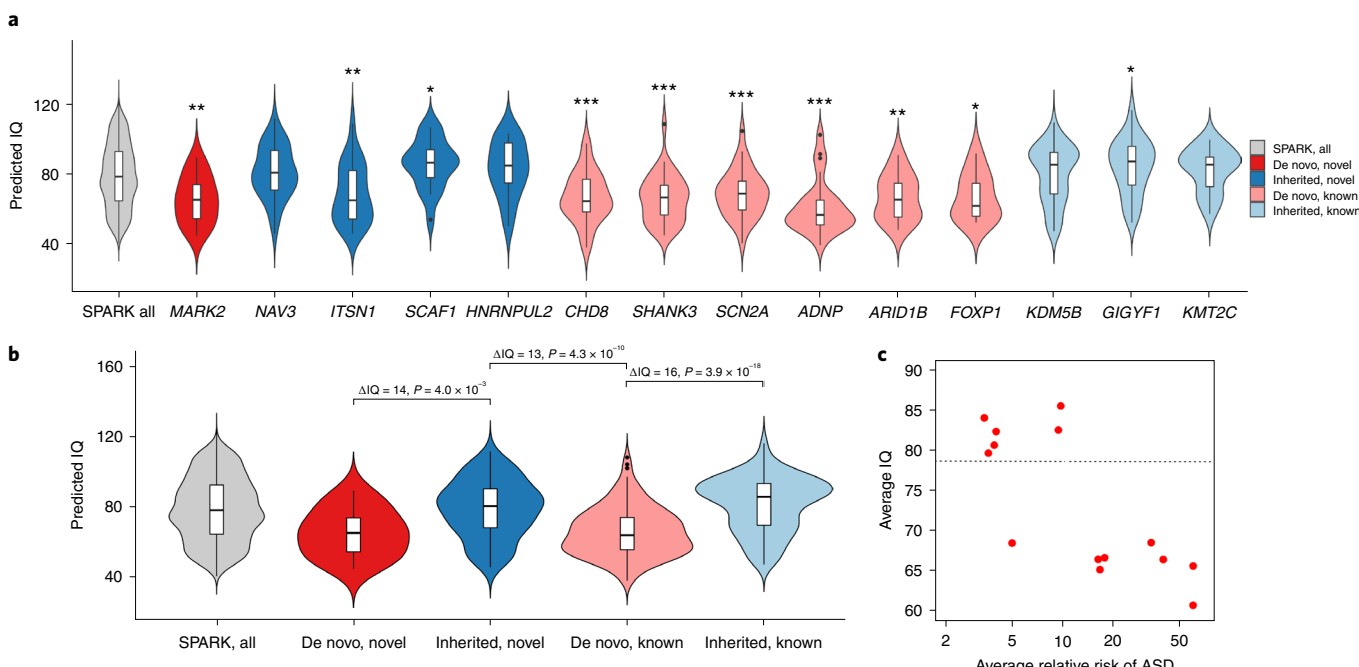

**Fig. 6 | Predicted full-scale IQ in individuals with pathogenic variants in inherited or de novo genes in SPARK.** We examined the distribution of predicted IQ using a machine learning method[49] for 95 individuals with ASD with an LoF mutation in one of the five novel exome-wide-significant genes (*MARK2*, *NAV3*, *ITSN1*, *SCAF1* and *HNRNPUL2*) and nine known ASD genes (*CHD8*, *SHANK3*, *SCN2A*, *ADNP*, *ARID1B*, *FOXP1*, *KDM5B*, *GIGYF1* and *KMT2C*), compared with 2,545 SPARK participants with ASD and known IQ scores. The nine known ASD genes include six genes (pink and labeled 'de novo, known') that are well-established de novo ASD risk genes that exceed exome-wide significance and were most frequently identified in SPARK, which maximizes the number of samples available for genotype–phenotype analyses. We also included three genes (light blue and labeled 'inherited, known') that have some previous evidence for inherited ASD risk (*GIGYF1*[7], *KDM5B*[62] and *KMT2C*[63]) and were also frequently identified in SPARK. We denote the genes contributing to ASD primarily through de novo LoF variants in our analysis as 'de novo' (red), and the genes primarily through inherited LoF variants as 'inherited' (blue). **a**, Distribution of predicted IQ between individuals with ASD with LoF mutations in the five novel genes, nine known genes and all participants with ASD and known IQ scores in SPARK (n = 2,545). We compared the mean predicted IQ between participants with LoF mutations in ASD genes and all participants by two-sample *t*-test. *, $0.01 \leq P < 0.05$; **, $0.001 \leq P < 0.01$; ***, $P < 0.001$. Exact *P* values are listed in Supplementary Table 19. The box plots represent median as center, and interquartile range (IQR) as bounds of the box; the upper whisker extends from the upper bound of the box to $1.5 \times IQR$, and the lower whisker extends from the lower bound of the box to $1.5 \times IQR$. Two-sided *t*-test was used to compute the *P* values for comparing mean predicted IQ between ASD individuals with LoF mutation in specific gene and all ASD participants. Individuals with pathogenic variants in de novo risk genes have significantly lower predicted IQ than overall SPARK participants with ASD and known IQ scores, whereas individuals with LoF variants in moderate-risk, inherited genes show similar predicted IQ as the overall SPARK participants, with the exception of *ITSN1*. **b**, Distribution of predicted IQ between individuals with ASD gene grouped by both inheritance status ('de novo' or 'inherited') and whether the ASD genes are novel ('novel' or 'known'). We compared the mean predicted IQ between individuals with pathogenic variants in de novo genes and inherited genes among our five novel genes and nine known genes. Overall, people with LoF mutations in de novo genes have an average of 13–16 points lower predicted IQ than individuals with LoF mutations in inherited genes, regardless of whether the ASD genes are novel or known. The box plots represent median as center, and IQR as the bounds of the box; the upper whisker extends from the upper bound of the box to $1.5 \times IQR$, and the lower whisker extends from the lower bound of the box to $1.5 \times IQR$. **c**, Average relative risk of ASD and average predicted IQ among different groups. Each dot shows the average of individuals with rare LoFs of a gene selected in **a**. The relative risk is estimated from mega-analysis and capped at 60. Pearson correlation between average IQ and log relative risk is −0.78 (P = 0.001). The horizontal line represents the average IQ (IQ = 79) of all SPARK individuals with predicted IQs. *ITSN1* is an outlier at the bottom left corner.

and about 64,000 cases (twice the current sample size) to find genes with similar effect sizes as *SCAF1* (relative risk = 8) and *HNRNPUL2* (relative risk = 9). We note that ten and five times the current sample size, respectively, are required to detect genes similar to *NAV3* or *ITSN1* and genes similar to *SCAF1* or *HNRNPUL2* by DNVs alone.

## Discussion

In this study, we identified five new ASD risk genes by both DNVs and rare inherited coding variants. We identified rare LoF variants in new ASD risk genes with modest effect size that are not strongly associated with intellectual disability. This finding represents a difference in phenotypic association with intellectual disability compared with other highly penetrant ASD genes. To find new risk genes with relative risks of 2–5 (comparable to the low relative risk genes from this study, *NAV3* and *ITSN1*) in the 50th percentile for

gene-wide LoF mutation rate ($2 \times 10^{-6}$) and the 50th percentile for selection among known risk genes (0.2), our power analysis suggests that 52,000, 73,000, 116,000 or 227,000 total ASD cases are necessary, respectively (cf. equation (1) from power calculation in Supplementary material).

Our results suggest that the identification of new risk genes with rare inherited variants may substantially improve genetic diagnostic yield. We found that rare inherited LoF variants account for 6% of PAR, similar to de novo LoF variants. Over two-thirds of the PAR from coding DNVs is explained by known ASD or NDD genes. In contrast, less than 20% of PAR from rare inherited LoFs variants is explained by known genes, suggesting that most genes contributing to ASD risk through rare inherited variants are yet to be discovered. These unknown risk genes are still largely constrained to LoFs in the general population and/or have similar expression profiles in

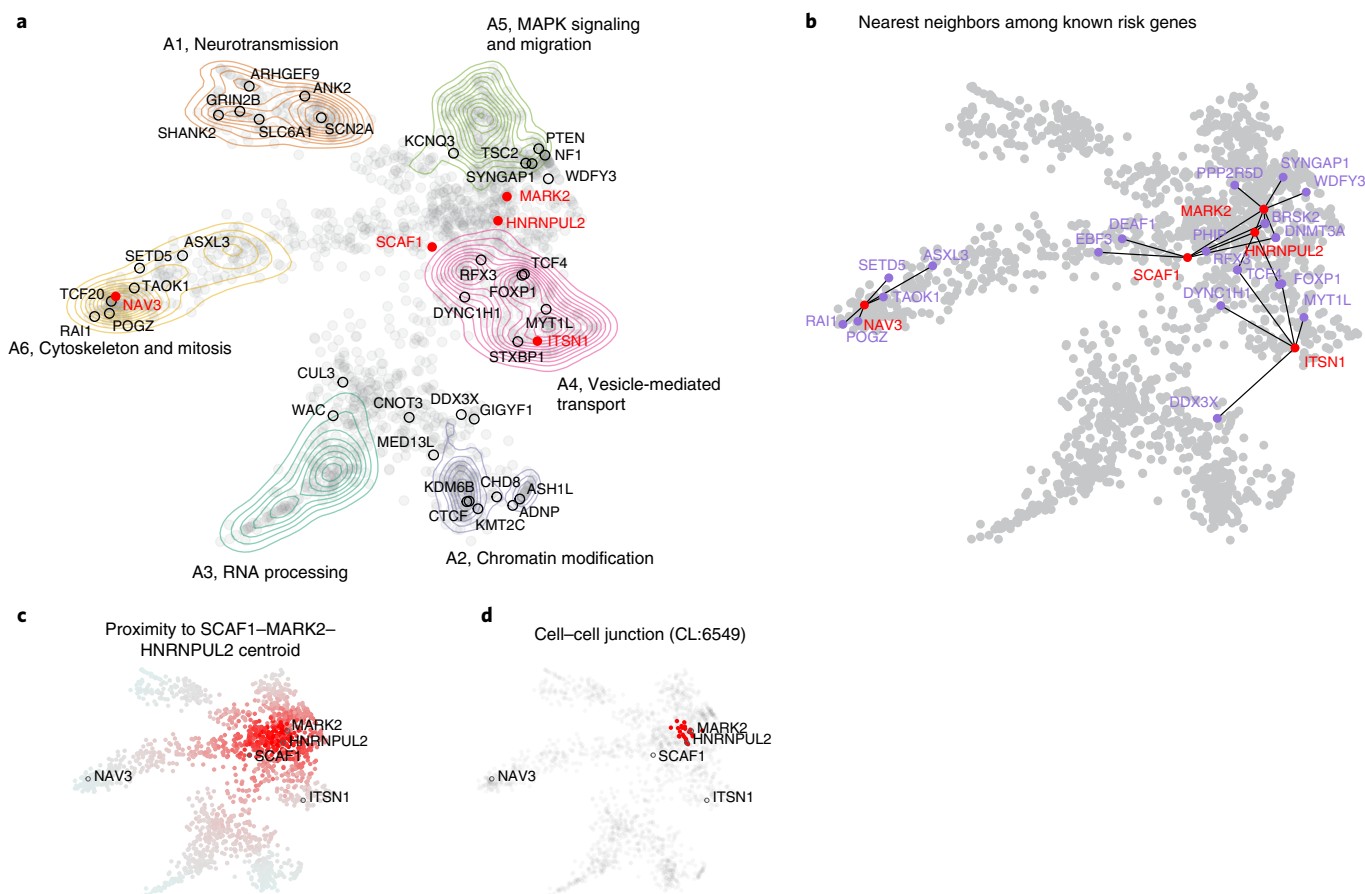

**Fig. 7 | Functional or phenotypic embedding of ASD risk genes. a**, Using a combination of archetypal analysis and canonical correlation analysis, putative autism risk genes were organized into $k = 6$ archetypes that represent distinct mechanistic (STRING) and phenotypic (HPO) categorizations (neurotransmission, chromatin modification, RNA processing, vesicle-mediated transport, MAPK signaling and migration, and cytoskeleton and mitosis). Genes implicated by our meta-analysis are indicated by their label, with novel genes in red. **b**, For each of the five novel genes, we identified the five nearest neighbors in the embedding space among the 62 meta-analysis genes. *SCAF1*, *MARK2* and *HNRNPUL2* were identified as 'mixed' rather than 'archetypal' in their probable risk mechanisms. **c**, To gain further insight into possible risk mechanisms, we calculated the embedding distance to the centroid of these three genes, which was then used as an index variable to perform gene set enrichment analysis. **d**, A STRING cluster (CL:6549) containing genes related to cell–cell junctions and the gap junction was identified as being highly localized in this region of the embedding space ($P = 4.12 \times 10^{-14}$ by the Kolmogorov–Smirnov test). This may suggest that these genes confer autism risk through dysregulation of processes related to cell adhesion and migration.

developing brains to known ASD risk genes. Combining evidence from both DNVs and rare inherited variants, we identified 60 genes associated with ASD with exome-wide significance, including five novel genes. Rare LoFs in these five new genes account for a PAR of 0.27%, about half of the PAR of the five most common highly penetrant ASD genes (*KDM5B*, *GIGYF1*, *CHD8*, *SCN2A* and *SHANK3*).

*NAV3*'s association to autism is primarily driven by rare inherited variants. Carriers of rare LoFs in *NAV3* have an average predicted IQ of 81, slightly above the SPARK cohort average (IQ, 79). The prevalence of intellectual disability among *NAV3* heterozygotes is similar to the SPARK cohort average. This is distinctly different from established ASD risk genes (for example, *CHD8*, *SHANK3* and *SCN2A*), nearly all identified by highly penetrant DNVs, and are associated with intellectual disability in ASD cohorts[2]. The absence of intellectual disability is also observed in other genes (for example, *SCAF1*, *HNRNPUL2*, *GIGYF1*, *KDM5B* and *KMT2C*) with substantial contribution from rare inherited variants and modest effect size. Nevertheless, the data show that many individuals with variants in these new ASD genes are affected with various neuropsychiatric conditions such as epilepsy, schizophrenia, Tourette syndrome and attention deficit hyperactivity disorder (ADHD) (Supplementary Table 12).

Detailed phenotyping of larger numbers of individuals carrying these rare inherited variants is needed to understand the phenotype associated with each gene. Such strategies should include a genetic and phenotypic assessment of family members who also carry the rare variant without ASD. Because all individuals consented in SPARK are recontactable, such studies will enable a more complete picture of the broad phenotypic effects of these variants without the bias of clinical ascertainment. Overall, these risk genes with modest effect size may represent a different class of ASD genes that are more directly associated with core symptoms of ASD and/or neuropsychiatric conditions rather than global brain development and intellectual disability.

The approaches used in this study made full use of rare variation, and this analytical method is generalizable to many conditions. In particular, the multiple methods used to reduce noise in LoF alleles present in control samples were particularly effective in assessing the signal within the novel genes of moderate effect. We also leveraged gene expression profiles informed by machine learning methods to help prioritize genes for the meta-analysis stage of our analysis[38]. Future studies that leverage additional multiomic data such as the Genotype–Tissue Expression (GTEx) project may further improve the signal-to-noise ratio.

Our archetypal analysis (Supplementary Tables 13–15) provides some clues to the potential mechanisms of the five newly identified risk genes. *ITSN1* was unambiguously mapped to A4 (vesicle-mediated transport) and has a role in coordinating endocytic membrane traffic with the actin cytoskeleton[57,58]. *NAV3* (A6, cytoskeleton and mitosis) is associated with both axon guidance[59] and malignant growth and invasion[60] and is thought to regulate cytoskeletal dynamics. Indeed, A6 is enriched for processes related to intermediate filaments (Supplementary Table 14), a known determinant of cell motility and polarity[61]. Although *MARK2*, *SCAF1* and *HNRNPUL2* were not identified as archetypal (potentially suggesting divergence from well-known autism risk mechanisms), a search for functional enrichment of this interstitial region between A4 and A5 found that their roles in developmental risk may be most relevant at the cell–cell junction, particularly as it relates to migration (see Fig. 7d).

Taken together, our results suggest that a continued focus on DNVs for ASD gene discovery may yield diminishing returns. By contrast, studies designed to identify genomic risk from rare and common inherited variants will not only yield new mechanistic insight, but also help explain the high heritability of ASD. SPARK is designed to recruit individuals across the autism spectrum, without relying on ascertainment at medical centers. As a result, SPARK may be better suited to identify genes with transmitted variants that have lower penetrance and to identify the genetic contributions to the full spectrum of autism. The strategy used by SPARK—to recruit and assess large numbers of individuals with autism across the spectrum and their available family members without costly, in-depth clinical phenotyping—is necessary to achieve the required sample size to fully elucidate genetic contributions to ASD. The ability to recontact and follow all SPARK participants will also be critical to deeply assess the phenotypes associated with the newly discovered genes and to develop and test novel treatments.

## Online content

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

## The SPARK Consortium

Adrienne Adams[14], Alpha Amatya[3], Alicia Andrus[15], Asif Bashar[3], Anna Berman[16], Alison Brown[17], Alexies Camba[18], Amanda C. Gulsrud[18], Anthony D. Krentz[19], Amanda D. Shocklee[20], Amy Esler[21], Alex E. Lash[3], Anne Fanta[22], Ali Fatemi[23], Angela Fish[24], Alexandra Goler[3], Antonio Gonzalez[25], Anibal Gutierrez Jr.[25], Antonio Hardan[26], Amy Hess[27], Anna Hirshman[14], Alison Holbrook[3], Andrea J. Ace[3], Anthony J. Griswold[28], Angela J. Gruber[19], Andrea Jarratt[29], Anna Jelinek[30], Alissa Jorgenson[29], A. Pablo Juarez[31], Annes Kim[24], Alex Kitaygorodsky[2], Addie Luo[32], Angela L. Rachubinski[33], Allison L. Wainer[14], Amy M. Daniels[3], Anup Mankar[3], Andrew Mason[34], Alexandra Miceli[16], Anna Milliken[35], Amy Morales-Lara[36], Alexandra N. Stephens[3], Ai Nhu Nguyen[3], Amy Nicholson[31], Anna Marie Paolicelli[37], Alexander P. McKenzie[17], Abha R. Gupta[22], Ashley Raven[24], Anna Rhea[38], Andrea Simon[39], Aubrie Soucy[40], Amy Swanson[16], Anthony Sziklay[34], Amber Tallbull[33], Angela Tesng[29], Audrey Ward[38], Allyson Zick[24], Brittani A. Hilscher[41], Brandi Bell[38], Barbara Enright[42], Beverly E. Robertson[3], Brenda Hauf[43], Bill Jensen[3], Brandon Lobisi[25], Brianna M. Vernoia[3], Brady Schwind[3], Bonnie VanMetre[17], Craig A. Erickson[30], Catherine A. W. Sullivan[22], Charles Albright[27], Claudine Anglo[41], Cate Buescher[8], Catherine C. Bradley[38], Claudia Campo-Soria[29], Cheryl Cohen[3], Costanza Colombi[24], Chris Diggins[3], Catherine Edmonson[17], Catherine E. Rice[44], Carrie Fassler[30], Catherine Gray[43], Chris Gunter[45], Corrie H. Walston[43], Cheryl Klaiman[45], Caroline Leonczyk[14], Christa Lese Martin[46], Catherine Lord[18], Cora M. Taylor[46], Caitlin McCarthy[38], Cesar Ochoa-Lubinoff[47], Crissy Ortiz[38], Cynthia Pierre[14], Cordelia R. Rosenberg[33], Chris Rigby[3], Casey Roche[38], Clara Shrier[38], Chris Smith[34], Candace Van Wade[38], Casey White-Lehman[3], Christopher Zaro[35], Cindy Zha[25], Dawn Bentley[15], Dahriana Correa[25], Dustin E. Sarver[48], David Giancarla[25], David G. Amaral[41], Dain Howes[29], Dalia Istephanous[29], Daniel Lee Coury[27], Deana Li[41], Danica Limon[39], Desi Limpoco[32], Diamond Phillips[14], Desiree Rambeck[29], Daniela Rojas[26],

Diksha Srishyla[29], Danielle Stamps[48], Dennis Vasquez Montes[3], Daniel Cho[49], Dave Cho[3],
Emily A. Fox[49], Ethan Bahl[8], Elizabeth Berry-Kravis[47], Elizabeth Blank[30], Erin Bower[34],
Elizabeth Brooks[3], Eric Courchesne[34], Emily Dillon[17], Erin Doyle[38], Erin Given[27], Ellen Grimes[16],
Erica Jones[3], Eric J. Fombonne[32], Elizabeth Kryszak[27], Ericka L. Wodka[17], Elena Lamarche[43],
Erica Lampert[30], Eric M. Butter[27], Eirene O'Connor[3], Edith Ocampo[14], Elizabeth Orrick[26],
Esmeralda Perez[3], Elizabeth Ruzzo[9], Emily Singer[3], Emily T. Matthews[35], Ernest V. Pedapati[30],
Faris Fazal[32], Fiona K. Miller[24], Gabriella Aberbach[35], Gabriele Baraghoshi[15], Gabrielle Duhon[39],
Gregory Hooks[29], Gregory J. Fischer[19], Gabriela Marzano[39], Gregory Schoonover[15],
Gabriel S. Dichter[43], Gabrielle Tiede[27], Hannah Cottrell[20], Hannah E. Kaplan[34], Haidar Ghina[49],
Hanna Hutter[17], Hope Koene[22], Hoa Lam Schneider[25], Holly Lechniak[14], Hai Li[47], Hadley Morotti[32],
Hongjian Qi[34], Harper Richardson[38], Hana Zaydens[3], Haicang Zhang[34], Haoquan Zhao[34],
Ivette Arriaga[18], Ivy F. Tso[50], John Acampado[3], Jennifer A. Gerdts[49], Josh Beeson[47], Jennylyn Brown[3],
Joaquin Comitre[25], Jeanette Cordova[33], Jennifer Delaporte[20], Joseph F. Cubells[44], Jill F. Harris[42],
Jared Gong[26], Jaclyn Gunderson[29], Jessica Hernandez[3], Jessyca Judge[24], Jane Jurayj[22], J. Kiely Law[3],
Julie Manoharan[3], Jessie Montezuma[38], Jason Neely[17], Jessica Orobio[39], Juhi Pandey[51], Joseph Piven[43],
Jose Polanco[30], Jibrielle Polite[3], Jacob Rosewater[25], Jessica Scherr[27], James S. Sutcliffe[52],
James T. McCracken[18], Jennifer Tjernagel[3], Jaimie Toroney[3], Jeremy Veenstra-Vanderweele[53],
Jiayao Wang[34], Katie Ahlers[49], Kathryn A. Schweers[14], Kelli Baalman[39], Katie Beard[29],
Kristen Callahan[48], Kendra Coleman[34], Kate D. Fitzgerald[24], Kate Dent[46], Katharine Diehl[3],
Kelsey Gonring[47], Katherine G. Pawlowski[35], Kathy Hirst[20], Karen L. Pierce[34], Karla Murillo[18],
Kailey Murray[43], Kerri Nowell[20], Kaela O'Brien[30], Katrina Pama[17], Kelli Real[43], Kaitlyn Singer[46],
Kaitlin Smith[41], Kevin Stephenson[27], Katherine Tsai[18], Leonard Abbeduto[41], Lindsey A. Cartner[3],
Landon Beeson[32], Laura Carpenter[38], Lucas Casten[8], Leigh Coppola[32], Lisa Cordiero[33],
Lindsey DeMarco[51], Lillian D. Pacheco[32], Lorena Ferreira Corzo[47], Lisa H. Shulman[36],
Lauren Kasperson Walsh[46], Laurie Lesher[15], Lynette M. Herbert[25], Lisa M. Prock[35], Lacy Malloch[48],
Lori Mann[3], Luke P. Grosvenor[3], Laura Simon[29], Latha V. Soorya[14], Lucy Wasserburg[29], Lisa Yeh[14],
Lark Y. Huang-Storms[32], Michael Alessandri[25], Marc A. Popp[19], Melissa Baer[47], Malia Beckwith[42],
Myriam Casseus[42], Michelle Coughlin[35], Mary Currin[43], Michele Cutri[25], Malcolm D. Mallardi[3],
Megan DuBois[29], Megan Dunlevy[45], Martin E. Butler[3], Margot Frayne[26], McLeod F. Gwynette[54],
Mohammad Ghaziuddin[24], Monica Haley[18], Michelle Heyman[37], Margaret Hojlo[35], Michelle Jordy[43],
Michael J. Morrier[44], Misia Kowanda[3], Melinda Koza[17], Marilyn Lopez[42], Megan McTaggart[17],
Megan Norris[27], Melissa N. Hale[25], Molly O'Neil[36], Madison Printen[14], Madelyn Rayos[25],
Mahfuza Sabiha[3], Mustafa Sahin[55], Marina Sarris[3], Mojeeb Shir[34], Matthew Siegel[56], Morgan Steele[26],
Megan Sweeney[20], Maira Tafolla[18], Maria Valicenti-McDermott[36], Mary Verdi[56], Megan Y. Dennis[57],
Nicolas Alvarez[17], Nicole Bardett[16], Natalie Berger[14], Norma Calderon[14], Nickelle Decius[25],
Natalia Gonzalez[42], Nina Harris[16], Noah Lawson[3], Natasha Lillie[29], Nathan Lo[3], Nancy Long[27],
Nicole M. Russo-Ponsaran[14], Natalie Madi[30], Nicole Mccoy[30], Natalie Nagpal[3], Nicki Rodriguez[41],
Nicholas Russell[27], Neelay Shah[3], Nicole Takahashi[20], Nicole Targalia[33], Olivia Newman[29],
Opal Y. Ousley[44], Peter Heydemann[47], Patricia Manning[30], Paul S. Carbone[15], Raphael A. Bernier[49],
Rachel A. Gordon[14], Rebecca C. Shaffer[30], Robert D. Annett[48], Renee D. Clark[43], Roger Jou[22],
Rebecca J. Landa[17], Rachel K. Earl[49], Robin Libove[26], Richard Marini[3], Ryan N. Doan[40],
Robin P. Goin-Kochel[39], Rishiraj Rana[3], Richard Remington[3], Roman Shikov[17], Robert T. Schultz[51],
Shelley Aberle[32], Shelby Birdwell[20], Sarah Boland[22], Stephanie Booker[30], S. Carpenter[16],

Sharmista Chintalapalli[24], Sarah Conyers[38], Sophia D'Ambrosi[38], Sara Eldred[27], Sunday Francis[29], Swami Ganesan[3], Susan Hepburn[33], Susannah Horner[51], Samantha Hunter[20], Stephanie J. Brewster[58], Soo J. Lee[14], Suma Jacob[29], Stanley Jean[3], So Hyun[59], Sydney Kramer[8], Sandra L. Friedman[33], Sarely Licona[14], Sandy Littlefield[26], Stephen M. Kanne[20,60], Sarah Mastel[32], Sheena Mathai[45], Sophia Melnyk[30], Sarah Michaels[17], Sarah Mohiuddin[24], Samiza Palmer[51], Samantha Plate[51], Shanping Qiu[37], Shelley Randall[30], Sophia Sandhu[18], Susan Santangelo[56], Swapnil Shah[3], Steve Skinner[61], Samantha Thompson[41], Sabrina White[48], Stormi White[45], Sabrina Xiao[3], Sidi Xu[34], Simon Xu[3], Tia Chen[34], Tunisia Greene[3], Theodore Ho[49], Teresa Ibanez[27], Tanner Koomar[8], Tiziano Pramparo[34], Tara Rutter[49], Tamim Shaikh[33], Thao Tran[47], Timothy W. Yu[40], Virginia Galbraith[38], Vahid Gazestani[62], Vincent J. Myers[3], Vaikunt Ranganathan[51], Vini Singh[17], William Curtis Weaver[46], Wenteng Cal[29], Wubin Chin[3], Wha S. Yang[18], Y. B. Choi[59] and Zachary E. Warren[31]

[14]Department of Psychiatry and Behavioral Sciences, Rush University Medical Center, Chicago, IL, USA. [15]Department of Pediatrics, University of Utah, Salt Lake City, UT, USA. [16]Vanderbilt Kennedy Center, Vanderbilt University Medical Center, Nashville, TN, USA. [17]Center for Autism and Related Disorders, Kennedy Krieger Institute, Baltimore, MD, USA. [18]Department of Psychiatry and Biobehavioral Sciences, University of California, Los Angeles, Los Angeles, CA, USA. [19]PreventionGenetics, Marshfield, WI, USA. [20]Thompson Center for Autism and Neurodevelopmental Disorders, University of Missouri, Columbia, MO, USA. [21]Department of Pediatrics, University of Minnesota, Minneapolis, MN, USA. [22]Child Study Center, Yale School of Medicine, New Haven, CT, USA. [23]Department of Neurogenetics, Kennedy Krieger Institute, Baltimore, MD, USA. [24]Department of Psychiatry, University of Michigan, Ann Arbor, MI, USA. [25]Department of Psychology, University of Miami's Center for Autism and Related Disabilities (UM-CARD), Coral Gables, FL, USA. [26]Department of Psychiatry and Behavioral Sciences, Stanford University, Stanford, CA, USA. [27]Division of Pediatric Psychology and Neuropsychology, Nationwide Children's Hospital (Child Development Center), Columbus, OH, USA. [28]John P. Hussman Institute for Human Genomics, University of Miami Miller School of Medicine, Miami, FL, USA. [29]Department of Psychiatry, University of Minnesota, Minneapolis, MN, USA. [30]Department of Psychiatry and Behavioral Neuroscience, Cincinnati Children's Hospital Medical Center - Research Foundation, Cincinnati, OH, USA. [31]Department of Pediatrics, Vanderbilt University Medical Center, Nashville, TN, USA. [32]Department of Psychiatry, Oregon Health & Science University, Portland, OR, USA. [33]Department of Pediatrics, JFK Partners/University of Colorado School of Medicine, Aurora, CO, USA. [34]Department of Neurosciences, University of California, San Diego and SARRC Phoenix, La Jolla, CA, USA. [35]Department of Pediatrics, Boston Children's Hospital, Boston, MA, USA. [36]Department of Pediatrics, Montefiore Medical Center and The Albert Einstein College of Medicine, Bronx, NY, USA. [37]Department of Psychiatry, Weill Cornell Medicine, White Plains, NY, USA. [38]Department of Pediatrics, Medical University of South Carolina, Charleston, SC, USA. [39]Department of Pediatrics, Texas Children's Hospital (Baylor College of Medicine), Houston, TX, USA. [40]Department of Medicine, Boston Children's Hospital, Boston, MA, USA. [41]MIND Institute and Department of Psychiatry and Behavioral Sciences, University of California, Davis, Sacramento, CA, USA. [42]Children's Specialized Hospital, Toms River, NJ, USA. [43]Department of Psychiatry, University of North Carolina (UNC, TEACCH, CIDD), Chapel Hill, NC, USA. [44]Department of Psychiatry and Behavioral Sciences, Emory University and Marcus Autism Center, Atlanta, GA, USA. [45]Department of Pediatrics, Emory University and Marcus Autism Center, Atlanta, GA, USA. [46]Geisinger Autism & Developmental Medicine Institute, Lewisburg, PA, USA. [47]Department of Pediatrics, Rush University Medical Center, Chicago, IL, USA. [48]Department of Pediatrics, University of Mississippi Medical Center, Jackson, MS, USA. [49]Department of Psychiatry and Behavioral Sciences, University of Washington/Seattle Children's Autism Center, Seattle, WA, USA. [50]Department of Psychology, University of Michigan, Ann Arbor, MI, USA. [51]Center for Autism Research, Children's Hospital of Philadelphia, Philadelphia, PA, USA. [52]Department of Molecular Physiology and Biophysics, Vanderbilt University, Nashville, TN, USA. [53]Department of Psychiatry, Columbia University Medical Center, New York, NY, USA. [54]Department of Psychiatry and Behavioral Sciences, Medical University of South Carolina, Charleston, SC, USA. [55]Department of Neurology, Boston Children's Hospital, Boston, MA, USA. [56]Maine Medical Center Research Institute, Scarborough, ME, USA. [57]Genome Center, MIND Institute, Department of Biochemistry and Molecular Medicine, University of California, Davis, Sacramento, CA, USA. [58]Translational Neuroscience Center, Boston Children's Hospital, Boston, MA, USA. [59]Center for Autism and the Developing Brain (CADB), Weill Cornell Medicine, White Plains, NY, USA. [60]Department of Health Psychology, University of Missouri, Columbia, MO, USA. [61]Greenwood Genetic Center, Greenwood, SC, USA. [62]Department of Pediatrics, University of California, San Diego and SARRC Phoenix, La Jolla, CA, USA.

## Methods

We established the SPARK cohort to facilitate genotype-driven research of ASD at scale[23]. All participants were recruited to SPARK under a centralized institutional review board (IRB) protocol (Western IRB Protocol no. 20151664). All participants provided written informed consent to take part in the study. Written informed consent was obtained from all legal guardians or parents for all participants aged 18 and younger and for all participants aged 18 and older who have a legal guardian. Assent was also obtained from dependent participants aged 10 and older. Participants with autism were compensated US$25–50 depending on other registered family members. The mean age of the SPARK cohort in this analysis was 16.5 years (s.d., 19.2 years). Cases made up 46% of the SPARK cohort (the remaining 54% were controls). The sex breakdown of the full SPARK cohort was 57% male and 43% female. The sex breakdown of the SPARK case cohort was 77% male and 23% female.

The first stage of analysis included 28,649 SPARK participants, including 10,242 ASD cases from over 9,000 families with exome sequencing data that passed quality control (Supplementary Data 1). A subset of 1,379 individuals was part of the published pilot study[7]. To replicate prioritized genes from the discovery stage, we performed a second-stage analysis that included an additional 39,926 individuals with 16,970 ASD cases from over 20,000 families with exome or WGS data available after the analysis in the discovery cohort was completed. New exome sequencing samples in this study were captured using an IDT xGen research panel and sequenced on Illumina NovaSeq. DNA samples were also genotyped for over 600,000 single nucleotide polymorphisms (SNPs) using Infinium Global Screening Array. Supplementary Table 16 outlines the software version and parameter settings for each analysis below. Details on data preprocessing, variant-level quality control procedures, variant annotation, high-confidence LoF variant filtering, and copy number variants, as well as descriptions of other publicly available ASD cohorts (SSC, MSSNG and ASC), are in the Supplementary Note.

**DNVs.** We identified candidate de novo single nucleotide variants (SNVs) or insertion–deletion mutations (indels) from SPARK and SSC cohorts from per-family variant call formats files (VCFs) generated by GATK[64] (v.4.1.2.0) and freebayes[65] (v.1.1.0) and a cohort-wide population VCF generated by weCall[66] (v.2.0.0) using a set of heuristic filters that aim to maximize the sensitivity while minimizing false negatives in parents[7] (Supplementary Table 16). Candidates were retained if they were called by DeepVariant[67] (v.0.8.0) in offspring and had no support in parents, identified in multiple offspring that passed the DeepVariant filter in all trios, or were shared by siblings in the same family and the de novo quality estimated by triodenovo was higher than 8 (or 7 for SNVs in CpG context). Before creating the final call set, we selected subsets of variants (see Supplementary Table 16) for manual evaluation by Integrative Genomics Viewer (IGV) to filter out candidates with failed review. Finally, we merged nearby clustered de novo coding variants (within 2 bp for SNVs or 50 bp for indels) on the same haplotype to form multinucleotide variants (MNVs) or complex indels. We removed variants located in regions known to be difficult for variant calling (HLA, mucin, and olfactory receptors). DNVs in the final call set follow a Poisson distribution with an average of 1.4 coding DNVs per affected offspring and 1.3 per unaffected offspring (Supplementary Fig. 15). The proportion of different types of DNVs, the mutation spectrum of SNVs, and indel length distributions were similar between SPARK and SSC (Supplementary Fig. 15). A small fraction of variants in the final call set are likely postzygotic mosaic mutations (Supplementary Fig. 16).

**Rare variants.** Rare variant genotypes were filtered from cohort-wide population VCFs with quality-control metrics collected from individual and family VCFs (Supplementary Fig. 17a). In brief, we initially extracted high-quality genotypes for each individual for variants that appear in less than 1% of families in the cohort. Evidence for the variant genotypes was re-evaluated by DeepVariant from aligned reads and collapsed over individuals to create site-level summary statistics including the fraction of individual genotypes that passed DeepVariant filter and mean genotype quality over all individuals. For variant genotypes extracted from GLnexus[68] (v1.1.3) VCFs, we re-examined variant genotype from per-family VCFs by GATK to collect GATK site-level metrics (including QD, MQ and SOR), then took the read-depth weighted average over families to create cohort-wide site metrics. For variant genotypes extracted from GATK joint genotyping VCFs, these site metrics were directly available from INFO fields.

**De novo analysis.** In the discovery-stage analysis, the DNV call sets of SPARK and SSC were merged with published DNVs from ASC[3,8] and MSSNG[6]. To infer likely sample overlaps with published trios for which we do not have individual-level data, we tallied the proportion of shared DNVs between all pairs of trios. For a pair of trios, let $N_1$ and $N_2$ be the number of coding DNVs, and let $O$ be the number of shared DNVs between the pair. To account for mutation hot spots, if a DNV is an SNV within CpG context or a known recurrent DNV identified in SPARK and SSC, it contributes 0.5 to the count. Likely overlapping samples were identified if ($O/N_1$) ≥ 0.5 or ($O/N_2$) ≥ 0.5 and they have identical sex.

To determine the expected number of DNVs in the cohort, we used a 7-mer mutation rate model[55] in which the expected haploid mutation rate of each base pair depends on the 3-bp sequence context on both sides. The per-base mutation

rates were adjusted by the fraction of callable trios at each base pair, which was the fraction of trios with ≥10× coverage in parents and ≥15× coverage in offspring. For published trios, we used an in-house WGS dataset of 300 trios (average, 36× coverage) to approximate the callable regions. Gene-level haploid mutation rates for different classes of DNVs were calculated by summing up the depth-adjusted per-base mutation rate of all possible SNVs of the same class. The rate for frameshift variants was presumed to be 1.3× the rate of stop-gained SNVs[56]. Mutation rates in haploid X chromosome regions were adjusted for the observed male:female ratio (4.2), assuming that mutation rates in spermatogenesis are 3.4 times higher than in oogenesis[9]. The exome-wide rate of synonymous DNVs closely matches the observed number of DNVs (Extended Data Fig. 1). We also observed similar fold enrichment of damaging DNVs (vs expected rate) in ASD cases across four cohorts after accounting for samples with family history (Extended Data Fig. 1).

To identify risk genes through DNVs, we applied DeNovoWEST[11]. We used the empirical burden of DNVs to derive weights for different variant classes in constrained genes (ExAC pLI ≥ 0.5) and nonconstrained genes separately based on positive predictive values (PPVs) (Supplementary Table 18). For ASD, we defined de novo D-mis variants by REVEL score ≥ 0.5. For other NDDs, we defined two classes of de novo D-mis variants by MPC score ≥ 2 or MPC ≥ 1 and CADD score ≥ 25. We first ran DeNovoWEST to test the enrichment of all nonsynonymous DNVs (pEnrichAll). To account for risk genes that harbor only missense variants, we ran DenovoWEST to test the enrichment of de novo missense variants only and applied a second test for spatial clustering of missense variants using denovonear[9], then combined evidence of missense enrichment and clustering (pCombMis). The minimal of pEnrichAll and pCombMis was used as the final $P$ value for DeNovoWEST. The exome-wide significance threshold was set to $1.3 \times 10^{-6}$ (=0.05/(18,000 genes × 2)). The analysis on the replication cohort used the same weights as derived from the discovery cohort. Compared with the original publication[11], our implementation of DeNovoWEST used different ways to stratify genes, determine variant weights and calculate per-base mutation rates. We applied our DeNovoWEST implementation on 31,058 NDD trios. The results are highly concordant with published results on the same data (Supplementary Fig. 18). We used $P$ values from our reanalysis on other NDD trios in comparative analysis with ASD.

Gene set enrichment analysis of DNVs was performed using the DNENRICH framework[32]. We included all de novo LoF and D-mis variants in 5,754 constrained genes from 16,877 ASD and 5,764 control trios. For each gene set, we calculated the fraction of weighted sums of damaging DNVs using PPV weights of constrained genes (Supplementary Table 18) for cases and controls respectively. The test statistic for each gene set is the ratio of such fractions in cases over controls. To determine the distribution of the test statistic under the null hypothesis, we randomly placed mutations onto the exome of all constrained genes, while holding the number of mutations, their trinucleotide context and functional impact to be the same as observed in cases and controls separately. Note that by conditioning on the observed number of damaging DNVs in cases and controls, we tested enriched gene sets in cases that are not due to an increased overall burden. At each round of simulation, the permuted test statistic in each gene set was calculated. Finally, the $P$ value was calculated as the number of times that the permuted statistic was no less than the observed statistic. Fold enrichment was calculated as the ratio between observed and average test statistics over all permutations. We also approximated the 95% confidence interval for fold enrichment by assuming that log[fold enrichment] follows normal distribution with mean 0 and s.d. determined by the $P$ value.

In all DNV analyses above, DNVs shared by full or twin siblings represent single mutational events and were counted only once. When an individual carries multiple DNVs within 100 bp in the same gene, only one variant with the most severe effects was included in the analysis.

**Transmission disequilibrium analysis.** The effect of inherited LoF variants was analyzed using TDT in each individual gene or in gene sets. Rare LoF variants were first identified in parents without ASD diagnoses or intellectual disability, then for each parent–offspring pair, the number of times the LoF variant was transmitted from parents to offspring was tallied. For variants in the (non-PAR part of the) X chromosome, we only used rare LoF variants carried by mothers without ASD diagnoses or intellectual disability and analyzed transmission in different types of mother–offspring pairs. For TDT analysis of rare inherited missense variants, different D-mis definitions and allele frequency cutoffs were used (Supplementary Fig. 2).

The overtransmission of LoFs to affected offspring was evaluated using a binomial test assuming that under the null hypothesis the chance of transmission is 0.5. In the discovery stage, ultra-rare LoFs with pExt ≥ 0.1 were used in exome-wide transmission disequilibrium and gene set enrichment analysis. For the per-gene test, all rare LoFs with pExt ≥ 0.1 were also used, and the TDT statistic[39] for each gene was calculated by $z = \frac{T - NT}{\sqrt{T + NT}}$, where $T(NT)$ is the number of times

LoF variants were transmitted (not transmitted) to affected offspring. When offspring included monozygotic twin pairs, only one was kept in the transmission analysis. We prioritized 244 autosomal genes with $z > 1$ in the top 10% of LOEUF or in the top 20% of LOEUF and A-risk ≥ 0.4. In the second stage gene-based test, if a gene-specific pExt threshold was available, we used high-confidence LoF variants that passed the gene-specific pExt filter (see Supplementary Note).

In the gene set enrichment analysis of inherited LoFs, the rate of transmission to affected offspring in each gene set was compared with the transmission rate in the rest of the genes in the background using the $\chi^2$ test.

**Case-control analysis.** Pseudocontrols are constructed from parents without ASD diagnoses or intellectual disability in simplex families, using alleles that were not transmitted to affected offspring. Each parent without ASD diagnoses or intellectual disability contributes a sample size of 0.5 to pseudocontrols. Rare LoFs in ASD cases whose parent data are not available and from other cases that were not used in DNV enrichment or TDT analysis are analyzed in this stage. Specifically, for each ASD case, we found out all of their most recent unaffected ancestors without ASD diagnoses or intellectual disability in the pedigree and calculated the contributing sample size as 1 minus the summation of kinship coefficients with these ancestors. If the contributing sample size was greater than 0, then the sample was included in pseudocases after removing alleles that were observed in any unaffected ancestors without ASD diagnoses or intellectual disability used in TDT and alleles included in DNV analysis if any. Examples of such rare LoFs in cases and their contributing sample sizes are given in Supplementary Fig. 19.

Rare LoFs in cases and controls for the X chromosome were categorized separately for males and females. Male controls include all fathers. In contrast, male cases include only those whose mothers do not have ASD diagnoses or intellectual disability (thus not included in TDT analysis). For females, because we include only mothers without ASD diagnoses or intellectual disability and affected sons in TDT, female pseudocases include all affected females. Female pseudocontrols were established from unaffected mothers in simplex families using alleles that do not transmit to affected sons. An unaffected mother contributes a sample size of 0.5 to pseudocontrols. In both sexes, DNVs were removed from pseudocases.

For gene-based case-control association in stage 2, we used population data as controls, including gnomAD exomes[26] (v2.1.1 non-neuro subset) and TOPMed genomes[69] (freeze 8). Variants in the population controls were filtered to keep those that passed the default quality-control filter in released data. For variants in gnomAD data, we further removed variants located in low-complexity regions[70] using the same procedure that was already applied to cases and TOPMed data. In gene-level case-control comparison of LoF burden, we used a baseline pExt ≥ 0.1 filter or gene-specific pExt threshold if available to define high-confidence LoF variants (see Supplementary Note). For LoF variants in selected genes, we also extracted curation results by gnomAD to remove curated non-LoF variants, and manually reviewed IGV snapshots from the gnomAD browser if available to remove likely variant calling artifacts (Supplementary Data 1). The number of high-confidence LoF variants was obtained from the allele counts in site-level VCF files. The gene-level burden of high-confidence LoF variants between cases and population controls is tested by comparing the high-confidence LoF variant rates between cases and controls using the Poisson test. The gene-level burden of ultra-rare synonymous variants (allele frequency $< 1 \times 10^{-5}$) between cases and population controls is assessed by a quantile–quantile (Q–Q) plot among all ancestry and European ancestry (Extended Data Fig. 3). To account for difference in depth of coverage, sample sizes are multiplied by the fraction of callable coding regions of each gene ($\geq 15 \times$ for autosomes or female X chromosome, $\geq 10 \times$ for male X chromosome) in ASD cases and in population controls respectively.

To account for sample relatedness in case-control analysis, we created a relationship graph in which each node represents an individual and each edge represents a known first-degree or second-degree relationship between two individuals. We also added edges to pairs of individuals without known familial relationship but have an estimated kinship coefficient ≥ 0.1. From the graph, we select one individual from each connected component to create unrelated case-control samples. For the X chromosome, father and sons were treated as unrelated. For population controls, only gnomAD data included sex-specific allele counts and were used in the sex-specific analysis.

Meta-analysis was performed for prioritized autosomal genes that are constrained (defined as ExAC pLI ≥ 0.5 or ranked in the top 20% of the LOEUF scores; Supplementary Data 2). We integrated evidence from the enrichment of all DNVs, transmission disequilibrium and increased burden in cases compared with population controls by combining P values using Fisher's method[40]. Study-wide significance was set at $9 \times 10^{-6}$ (Bonferroni correction with 5,754 constrained genes). In mega-analysis, we combined all unrelated ASD cases together and compared CAFs of high-confidence LoF variants with two population control sets.

**Power calculation.** To calculate statistical power of the current study and to estimate sample size for future studies, we adopted the statistical framework from ref. [41], comparing CAF of LoF variants in $N$ unrelated cases $f_{case}$ with CAF $f$ in the general population. The effects of LoFs in the same gene are assumed to be the same (relative risk = $\gamma$). We focus only on constrained genes. We assume $f$ to be at mutation–selection equilibrium $f = \mu_{LoF}/s$, where $\mu_{LoF}$ is the LoF mutation rate and $s$ is the selection coefficient. The test statistic asymptotically follows a noncentral $\chi^2$ distribution with 1 d.f. and noncentrality parameter (NCP):

$$\lambda = 4N \left[ \gamma f \ln \gamma + (1 - \gamma) \ln \frac{1 - \gamma f}{1 - f} \right]$$

Given the significance threshold $\alpha$, power can be calculated analytically by

$$1 - \beta = 1 - F\left[ F^{-1}(1 - \alpha, 0), \lambda \right]$$

where $F(x, \lambda)$ is the cumulative distribution of $\chi_1^2$ with NCP $\lambda$.

To calculate the sample size to achieve desired power $1 - \beta$ at significance level $\alpha$, we first solve NCP $\lambda_{\alpha,\beta}$ from the above equation, then estimate sample size by

$$n_{\alpha,\beta} \approx \frac{\lambda_{\alpha,\beta}}{4f[\gamma \ln \gamma - (\gamma - 1)]}$$

The current study has a sample size of $n = 31,976$ and the type I error rate is $\alpha = 9 \times 10^{-6}$ (experimental wide significant). Given continuing expansion of biobank-scale sequencing, treating $f$ as known without error is a reasonable assumption for future studies. To calculate power for the new genes identified in this study, we used point estimates of $\gamma$ and $f$ from mega-analysis using gnomAD exomes as population controls, and used $\mu_{LoF}$ computed from the 7-mer context-dependent mutation rate model[55] to convert $f$ to $s = \mu_{LoF}/f$. The required sample sizes were calculated to achieve 90% power.

Power and sample size are both calculated as a function of relative risk for ASD ($\gamma$) and selection coefficient ($s$) across different haploid LoF mutation rates ($\mu_{LoF}$). We only considered $s$ between 0.01 and 0.5, because most prioritized genes have point estimates of $s > 0.01$ (Supplementary Fig. 5) and genes with $s > 0.5$ are expected to harbor more de novo than inherited LoF variants and can be identified from the enrichment of DNVs. We limited relative risk ($\gamma$) between 1 and 20 to focus on risk genes with moderate to small effects. The reduction in fitness $s$ is correlated with the increases in ASD risk $\gamma$ by $s = \gamma \pi$ under the assumption of no pleiotropic effect, where $\pi$ is ASD prevalence and $s_D$ is decreased reproductive fitness of ASD cases. Based on epidemiological studies, the current estimated prevalence of ASD is $\hat{\pi} = 1/54$[71], and the estimated $s_D$ is for 0.75 male and for 0.52 female[72], so sex-averaged $\hat{s}_D = 0.71$ (assuming male:female ratio of 4.2). In reality, most known ASD genes also show pleiotropic effects with other NDDs or are associated with prenatal death; therefore, $s \geq \gamma \pi s_D \approx \gamma \hat{\pi} \hat{s}_D = 0.013\gamma$. So, we considered only combinations of $(s, \gamma)$ that satisfy the condition $s \geq 0.013\gamma$.

**Gene sets.** To evaluate the contribution of known ASD risk genes to the burdens of DNVs and inherited LoF variants identified in this study, we collected 618 known dominant ASD or NDD genes from the following sources: (1) Known NDD genes from DDG2P[73] (2020-02) that are dominant or X-linked and have an organ specificity list that includes the brain or cause multisystem syndrome; (2) high-confidence ASD genes collected by the Simons Foundation Autism Research Initiative (SFARI)[74] (2019-08) with score of 1 or 2, excluding known recessive genes; and (3) dominant ASD genes reported in recent literature and included in the SPARK genes list[75] (2020-07).

To evaluate the gene sets enriched by damaging DNVs or inherited high-confidence LoFs, we used all constrained genes by ExAC pLI ≥ 0.5 or in the top 20% of the LOEUF as the background. Gene sets of the following five categories were collected for gene set enrichment analysis.

*Transcriptome and proteome.* (1) Genes with brain-specific expression, defined as the genes with average reads per kilobase of transcript per million mapped reads (RPKM) > 1 in the brain and over four times the median RPKM of 27 tissues in processed RNA sequencing (RNA-seq) data from ref. [76]. (2) Genes in coexpression modules M2 and M3 derived from weighted gene correlation network analysis (WGCNA) of BrainSpan developmental RNA-seq data, previously reported to be enriched for known ASD genes[33]. (3) Genes expressed in excitatory or inhibitory neurons. We selected genes from ref. [77] that have average transcripts per million (TPM) > 100 in excitatory neurons and in inhibitory neurons. (4) Synaptic genes collected from SynaptomeDB[78].

*Neuronal regulome.* (1) Putative CELF4 target genes, defined as genes with an iCLIP occupancy > 0.2 in ref. [79]. (2) CHD8 target genes, defined as genes with promoter or enhancer regions that overlap with CHD8 binding peaks in human neural stem cells or midfetal brain in ref. [36]. (3) FMRP target genes[35] with a false discovery rate (FDR) < 0.1 that mapped them to orthologous human genes (Mouse Genome Informatics[80], 2018-07). (4) RBFOX2 target genes[34] that have Rbfox2 tag counts ≥ 8. Due to high correlations between RBFOX1 and RBFOX3, targeted genes by the two RNA binding proteins were merged in one gene set and selected to have total Rbfox1 and Rbfox3 tag counts > 24.

*Autism gene predictions.* (1) ForecASD is an ensemble classifier that integrates brain gene expression, heterogeneous network data and previous gene-level predictors of autism association to yield a single prediction score[37]. We created two sets of genes with forecASD prediction score greater than 0.4 or 0.5. (2) A-risk is a classifier that uses a gradient boosting tree to predict autism candidate genes using cell-type specific expression signatures in the fetal brain[38]. We created three sets of genes with prediction score greater than 0.4, 0.5 or 0.6.

*Genetic evidence.* (1) Genes enriched with DNVs in ASD, with nominal statistical evidence ($P < 0.01$ or $P < 0.05$ by DeNovoWEST) in the SPARK discovery cohort of

16,877 trios. (2) Genes enriched with DNVs in other NDDs (Supplementary Data 3), with nominal statistical evidence in 31,058 NDDs[11]. (3) Genes implicated in schizophrenia, with nominal statistical evidence ($P < 0.05$)[48].

*Genetic constraint.* Four constraint gene sets are defined by genes in top 10% of the gnomAD LOEUF, 10–20% the gnomAD LOEUF and genes with selection coefficient for heterozygous protein-truncating variants estimated from ExAC data (sHet) $\geq 0.25$ and $\geq 0.2$.

*Archetypal analysis.* Archetypal analysis[81] is an unsupervised learning approach that has similarities to other dimensionality reduction and clustering approaches. An important distinction of archetypal analysis from comparable approaches is that it seeks a set of $k$ archetypes, which are points along the convex hull of the data, from which all data points may be expressed as a mixture. The output of archetypal analysis is an $N$ by $k$ matrix, α, of [0,1] coefficients that represent the contribution of each archetype to each data point. Whereas cluster centroids are embedded within the interior of a cluster, archetypes are, by design, at the extremes of the data (that is, along the convex hull). In practice, this ensures that the archetypes are appreciably distinct from one another, which is often not the case for cluster centroids. Here, we use archetypal analysis as a means to organize putative ASD risk genes in a space that has meaning from a mechanistic (STRING) and phenotypic (HPO) standpoint. Genes of particular interest to this study can then be considered and interpreted in the context of the archetypes that define this space. STRING (v11)[52] clusters and HPO[53] terms were formatted as gene-by-term binary matrices. The working gene list was taken as the union of forecASD top decile genes and the 62 autism-associated genes from this study (total of 1,776 genes). A total of 583 genes from this set had annotations in both STRING and HPO, and using these genes, a canonical correlation analysis (CCA) was carried out using the RGCCA package for R (https://cran.r-project.org/web/packages/RGCCA/index.html) using five components and sparsity parameter c1 set to 0.8 for both the HPO and STRING matrices. Component scores for all 1,776 genes were calculated using the STRING cluster annotations and the corresponding coefficients from the CCA and was used as input for archetypal analysis[81], with the optimal $k$ (number of archetypes) selected using the elbow plot heuristic[82] and the residual sums of squares (RSS) plotted as a function of $k$. We displayed the archetypal embedding using the simplexplot() function of the archetypes R package. Genes were identified as 'archetypal' if their top archetype coefficient was >2× the next highest archetypal coefficient. Those genes that did not fulfill this criterion were classified as 'mixed', whereas those that did were assigned to their maximally scoring archetype. We applied the following heuristic to guide the naming of each archetype: each MSigDB[83] gene set (v7.4) that has two or more autism genes ($n = 62$ as described above) annotated is tested for association with the archetypal coefficients (A1–A6, simultaneously) in a quasibinomial generalized linear model (GLM) (Supplementary Table 15). The one-sided association *P* value (that is, positive association) for each archetype is reported in Supplementary Table 12 (top 20 for each archetype). To name each archetype, the top 20 gene sets with the strongest associations with that archetype were considered; this leads to the following naming conventions: A1, neurotransmission; A2, chromatin modification; A3, RNA processing; A4, vesicle-mediated transport; A5, MAPK signaling and migration; and A6, cytoskeleton and mitosis (see Supplementary Table 15 and Fig. 7). Representative genes for each archetype were chosen from among the list of 62 risk genes identified in this study, using the top six genes for each archetype (note that these genes do not necessarily fulfill the 'archetypal' criterion described above, but are simply the top six of the 62 for each archetype).

**Statistics and reproducibility.** No statistical method was used to predetermine sample size, and no data were excluded from the analyses.

**Reporting summary.** Further information on research design is available in the Nature Research Reporting Summary linked to this article.

## Data availability
In order to abide by the informed consents that individuals with autism and their family members signed when agreeing to participate in a SFARI cohort (SSC and SPARK), researchers must be approved by SFARI Base (https://base.sfari.org). To access to SPARK or SFARI data, researchers should
(1) Obtain a SFARI Base account at https://base.sfari.org, which will require affiliating with an institution. Currently, there are 271 institutions around the world that have signed SFARI's Researcher Distribution Agreement (RDA), and any researcher affiliated with those institutions can apply for SFARI Base access.
(2) Review the institute's executed RDA. The standard RDA is available at https://s3.amazonaws.com/sf-web-assets-prod/wp-content/uploads/sites/2/2021/06/15165956/SFARI_RDA.pdf
(3) Create a SFARI Base project, which includes a title, abstract and an IRB approval or exemption document.
(4) Create a SFARI Base request. All requests are processed in a timely manner. The SPARK data is accessible as follows:
SFARI_SPARK_iWES includes exome and genotyping data on 70,487 participants, including all people analyzed in this paper plus an additional 11,282 participants.

SFARI_SPARK_WGS_1 includes whole genome data from 2,629 individuals from 645 families with at least one person with autism.
SFARI_SPARK_WGS_2 includes whole genome data from 2,365 individuals from 587 families with at least one person with autism.
SFARI_SPARK_WGS_3 includes whole genome data from 2,871 individuals from 803 families with at least one person with autism.
SSC_WES_3 is whole exome data on the Simons Simplex Collection (SSC) as analyzed and reported in ref. [19].
SFARI_SSC_WGS_pilot contains genomes of 40 families with autism.
SFARI_SSC_WGS_1 and SFARI_SSC_ WGS_2 contain WGS of the SSC.
SSC Dataset contains phenotypic information on 2,644 simplex autism families.
SPARK Phenotype Dataset V7 is the current phenotypic dataset on 290,502 SPARK participants, including 111,720 participants with autism.

## Code availability
All software used in this study is publicly available. The code for major figures and analysis can be found at https://github.com/ShenLab/SPARK_Analysis_V1.git or https://doi.org/10.5281/zenodo.6646871.

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

## Acknowledgements
We are extremely grateful to the thousands of individuals and families who are participating in this study. We thank the sites, staff and volunteers of the SPARK Clinical Site Network and SFARI for their invaluable contributions. We are grateful to the many ASD advocacy and service organizations that have helped us inform the community

about SPARK, including the Autism Society of America and its affiliates, the Global and Regional Autism Spectrum Partnership (GRASP), Autism Speaks, and the Autism Science Foundation. We thank the members of SPARK's Community Advisory Council for providing feedback and advice. We thank members of our Scientific Advisory Board and SFARI scientists for advice on our protocol and participant outreach and retention strategies. Our de novo analyses involved primary data from SSC and results from published ASD studies, including ASC and MSSNG, and our case–control analysis used published population controls from gnomAD (exome v2.1.1, non-neuro subset) and TOPMed (WGS, freeze 8). Approved researchers can obtain the SPARK population dataset described in this study by applying at https://base.sfari.org. The SPARK initiative is funded by the Simons Foundation as part of SFARI. This research was supported, in part, by a grant from the National Institute of Mental Health (NIMH R01MH101221) and grants from the Simons Foundation (SFARI nos. 608045 and 810018EE) to E.E.E., a grant from the National Institutes of Health (NIH R01GM120609) and from the Simons Foundation (SFARI no. 606450) to Y.S., a grant from the Simons Foundation (SFARI no. 644038) to B.J.O. and J.J.M., a grant from the National Institute of Mental Health to T.N.T. (NIMH 1K99MH117165), and grants MH105527 and DC014489 from the National Institute of Health to J.J.M. E.E.E. is an investigator of the Howard Hughes Medical Institute.

## Author contributions

I. Astrovskaya, J.B.H., J.J.M., N.V., P.F., C. Shu, T.W., W.K.C., X.Z. and Y.S. designed and conceived this study. A. Adams, A. Andrus, A. Berman, A. Brown, A.C., A.C.G., A.D.S., A.E., A. Fanta, A. Fatemi, A. Fish, A. Goler, A. Gonzalez, A. Gutierrez Jr., A. Hardan, A. Hess, A. Hirshman, A. Holbrook, A.J.A., A.J. Griswold, A. Jarratt, A. Jelinek, A. Jorgenson, A. Juarez, A. Kim, A. Kitaygorodsky, A.L., A.L.R., A.L.W., A.M.D., A. Mankar, A. Mason, A. Miceli, A. Milliken, A.M.-L., A.N.S., A. Nguyen, A. Nicholson, A.P., A.P.M., A.R.G., A. Raven, A. Rhea, A. Simon, A. Swanson, A. Sziklay, A. Tallbull, A. Tesng, A.W., A.Z., B.A.H., B.B., B.E., B.E.R., B. Hauf, B.J.O., B.L., B.M.V., B.S., B.V., C.A.E., C.A.W.S., C. Albright, C. Anglo, C.B., C.C.B., C.C.-S., C. Cohen, C. Colombi, C.D., C.E., C.E.R., C. Fassler, C. Gray, C. Gunter, C.H.W., C.K., C. Leonczyk, C.L.M., C. Lord, C.M.T., C.M., C.O.-L., C. Ortiz, C.P., C.R.R., C. Roche, C. Shrier, C. Smith, C.V., C.W.-L., C. Zaro, C. Zha, D.B., Daniel Cho, D. Correa, D.E.S., D.G., D.G.A., D.H., D.I., D.L.C., D. Li, D. Limon, D. Limpoco, D.P., D. Rambeck, D. Rojas, D. Srishyla, D. Stamps, E.A.F., E. Bahl, E.B.-K., E. Blank, E. Bower, E. Brooks, E.C., E. Dillon, E. Doyle, E. Given, E. Grimes, E.J., E.J.F., E.K., E.L.W., E. Lamarche, E. Lampert, E.M.B., E. O'Connor, E. Ocampo, E. Orrick, E.P., E.R., E.S., E.T.M., E.V.P., F.F., F.K.M., G.A., G.B., G.D., G.H., G.M., G.S., G.S.D., G.T., H.C., H.E.K., H.G., H.H., H.K., H.L.S., H. Lechniak, H. Li, H.M., H.R., H. Zaydens, I. Arriaga, I.F.T., J.A., J.A.G., J. Beeson, J. Brown, J. Comitre, J. Cordova, J.D., J.F.C., J.F.H., J. Gong, J. Gunderson, J.H., J.J.M., J. Judge, J. Jurayj, J.K.L., J. Manoharan, J. Montezuma, J.N., J.O., J. Pandey, J. Piven, J. Polanco, J. Polite, J.R., J.R.W., J.S., J.S.S., J.T.M., J. Tjernagel, J. Toroney, J.V.-V., J.W., K.A., K.A.S., K. Baalman, K. Beard, K. Callahan, K. Coleman, K.D.F., K. Dent, K. Diehl, K.G., K.G.P., K.H., K.L.P., K. Murillo, K. Murray, K.N., K.O., K. Pama, K.R., K. Singer, K. Smith, K. Stephenson, K.T., L.A., L.A.C., L. Beeson, L. Carpenter, L. Casten, L. Coppola, L. Cordiero, L.D., L.D.P., L.F.C., L.G.S., L.H.S., L.K.W., L.L., L.M.H., L.M.P., L. Malloch, L. Mann, L.P.G., L.S., L.V.S., L.W., L.Y., L.Y.-H., M.A., M. Baer, M. Beckwith, M. Casseus, M. Coughlin, M. Currin, M. Cutri, M. DuBois, M. Dunlevy, M.F., M.F.G., M.G., M. Haley, M. Heyman, M. Hojlo, M.J., M.J.M., M. Kowanda, M. Koza, M.L., M.M., M.N., M.N.H., M.O., M.P., M.R., M. Sabiha, M. Sahin, M. Sarris, M. Shir, M. Siegel, M. Steele, M. Sweeney, M.T., M.V.-M., M. Verdi, M.Y.D., N.A., N. Bardett, N. Berger, N.C., N.D., N.G., N.H., N. Lillie, N. Long, N.M.R.-P., N. Madi, N. Mccoy, N.N., N. Rodriguez, N. Russell, N.S., N. Takahashi, N. Targalia, N.V., O.N., O.Y.O., P.F., P.H., P.M., P.S.C., R.A.B., R.A.G., R.C.S., R.D.A., R.D.C., R.J., R.J.L., R.K.E., R.L., R.P.G.-K., R. Remington, R.S., R.T.S., S.A., S. Birdwell, S. Boland, S. Booker, S. Carpenter, S. Chintalapalli, S. Conyers, S.D., S.D.B., S.E., S.F., S.G., S. Hepburn, S. Horner, S. Hunter, S.J.B., S.J.L., S. Jacob, S. Jean, S. Kim, S. Kramer, S.L.F., S. Licona, S. Littlefield, S.M.K., S. Mastel, S. Mathai, S. Melnyk, S. Michaels, S. Mohiuddin, S. Palmer, S. Plate, S.Q., S.R., S. Sandhu, S. Santangelo, S. Skinner, S.T., S. Xu, S. Xiao, Sabrina White, Stormi White, T.C., T.G., T.H., T.I., T.K., T.P., T.R., T.S., T.R.T., T.T., V. Galbraith, V. Gazestani, V.J.M., V.R., V.S., W.C.W., W. Cal, W.K.C., W.S.Y., Y.C. and Z.E.W. recruited participants and collected clinical data and biospecimens. A. Amatya, A. Bashar, A.E.L., A. Mankar, A. Nguyen, B.J., C. Rigby, Dave Cho, D.V.M., E. O'Connor, J.A., M.D.M., M.E.B., N. Lawson, N. Lo, N.V., R.M., R. Rana, S.G., S. Jean, S. Shah and W. Chin built and supported the SPARKforAutism.org website, software, databases and systems, and managed SPARK data. A.D.K., A.J. Gruber, A. Nishida, B. Han, B.J.O., C. Fleisch, C. Shu, D.V.M., E. Brooks, G.J.F., I. Astrovskaya, J.B.H., J.J.M., J.R.W., J.U.O., L. Brueggeman, L.G.S., M.A.P., N. Lo, N.V., O.M., P.F., S.D.B., S.C.M., S.X.X., T.N.T., T.S.C., T.W., W.T.H., X.Z. and Y.S. performed analyses, processed biospecimens and sequenced DNA samples. A. Fatemi, A. Kitaygorodsky, A. Soucy, C. Shu, D.H.G., E.B.-K., E.E.E., E.R., H.Q., H. Zhang, H. Zhao, I. Astrovskaya, J.B.H., J.J.M., J.R.W., J.U.O., L. Brueggeman, L.G.S., M.Y.D., N.V., O.M., P.F., R.N.D., S.D.B., S.C.M., S.X.X., T.K., T.N.T., T.P., T.S., T.R.T., T.W., T.W.Y., W.K.C., X.Z. and Y.S. helped with data interpretation. A.E.L., E.E.E., J.J.M., N.V., P.F., W.K.C. and X.Z. supervised the work. B.J.O., I. Astrovskaya, J.B.H., J.J.M., J.U.O., C. Shu, J.R.W., N.V., P.F., T.N.T., T.W., W.K.C., X.Z. and Y.S. wrote this paper.

## Competing interests

D.H.G. has received consulting fees or equity participation for scientific advisory board work from Ovid Therapeutics, Axial Biotherapeutics, Acurastem, and Falcon Computing. E.E.E. serves on the Scientific Advisory Board of Variant Bio. W.K.C. serves on Scientific Advisory Board of the Regeneron Genetics Center and is the Director of Clinical Research for SFARI. All other authors declare no competing interests.

## Additional information

**Extended data** is available for this paper at https://doi.org/10.1038/s41588-022-01148-2.

**Correspondence and requests for materials** should be addressed to Wendy K. Chung.

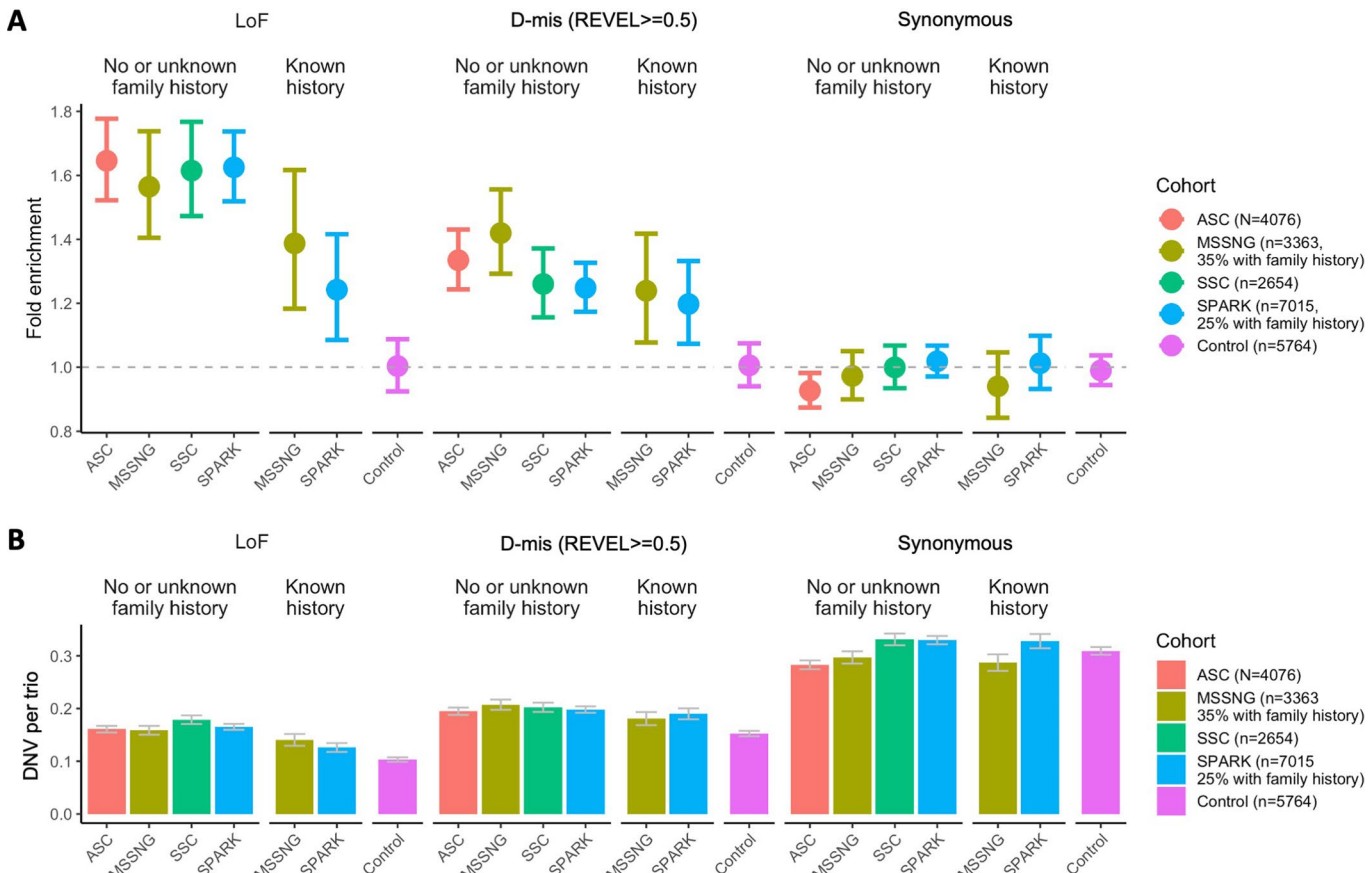

**Extended Data Fig. 1 | Overall burden of de novo variants in four ASD cohorts included in the discovery sample.** (A) Observed rates of de novo LoF, Dmis (REVEL > = 0.5) and silent variants (B) are compared with expected rates. We used a 7mer sequence context dependent mutation rate model[55] to calculate expected rates for different classes of de novo variants after adjusting sequencing coverage, and found a close match with observed de novo rates in control trios. The rates of de novo LoF and Dmis variants in ASD cases are significantly higher than baseline expectation and are reduced in cases with known family history. Data are presented as mean values and 95% confidence interval as error bars.

**A**   *De novo* LoF and D-mis variants     **B**   Transmission of rare, inherited LoFs

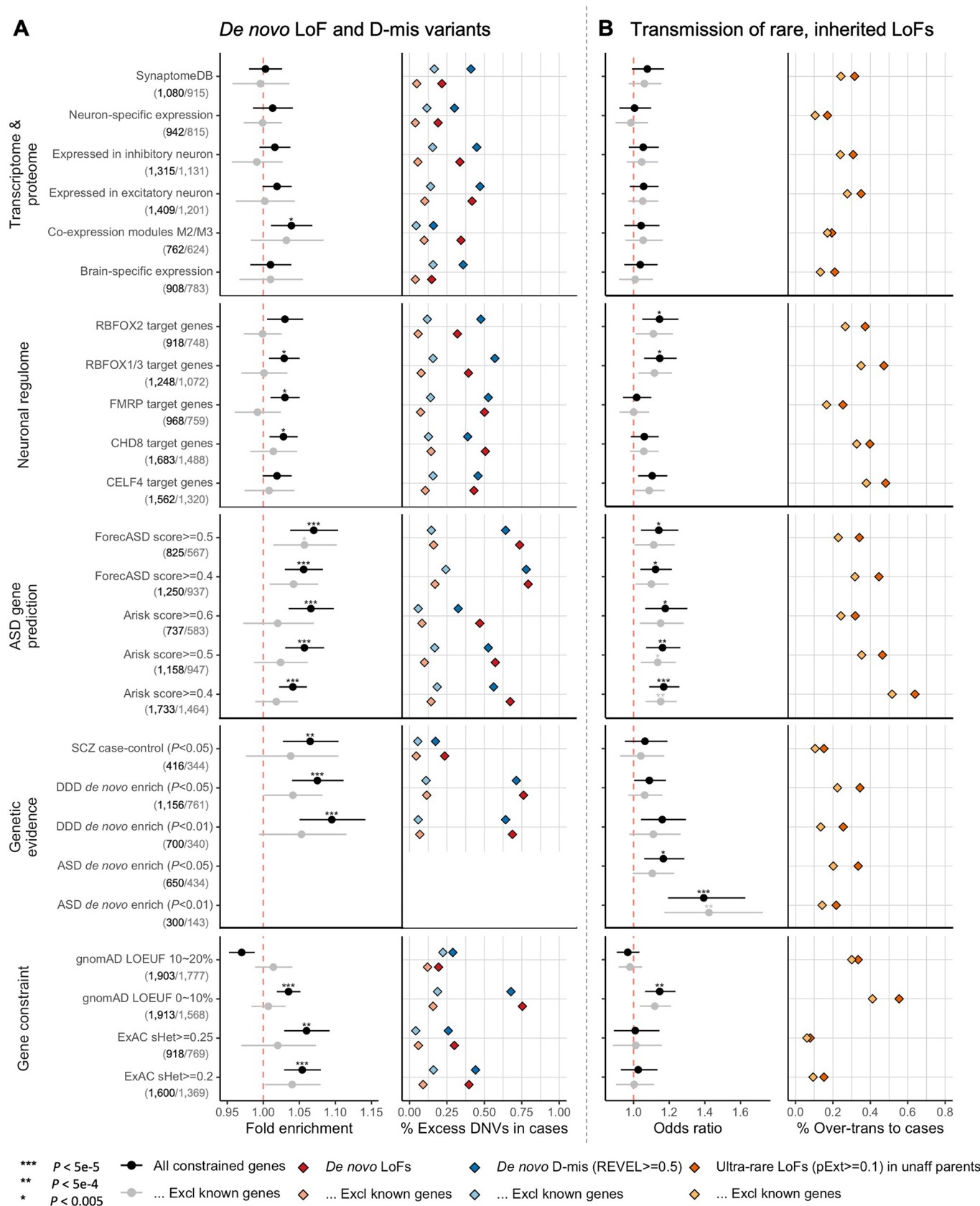

Extended Data Fig. 2 | See next page for caption.

 

**Extended Data Fig. 2 | Enrichment of de novo damaging and rare, inherited LoF variants in ASD cases across gene sets.** Gene sets were defined and grouped by transcriptome proteome, neuronal regulome, ASD gene prediction scores, genetic evidence from neuropsychiatric diseases, and gene level constraint. Analyses were repeated after removing known ASD/NDD genes. Number of genes in each set before and after removing known genes are shown in bracket below gene set. Dots represent fold enrichment of DNVs or odds ratio for over-transmission of LoFs in each set. Horizontal bars indicate 95% confidence interval. For each gene set, also shown are the percentage of excessive DNVs in cases and percentage of over-transmission of rare, inherited LoFs to cases. (A) De novo enrichment analysis was performed by dnEnrich that conditional on the overall increase in burden of de novo damaging variants in cases compared with controls (Methods). P-values were derived from 100,000 random permutations of de novo damaging variants among all 5,754 constrained genes and accounts for the tri-nucleotide sequence context and gene length. (B) Enrichment of rare, inherited LoFs was evaluated by comparing the transmission and non-transmission of ultra-rare LoFs with pExt >=0.1 in the gene set versus those in all other constrained genes using a 2-by-2 table. P-values were given by chi-squared test.

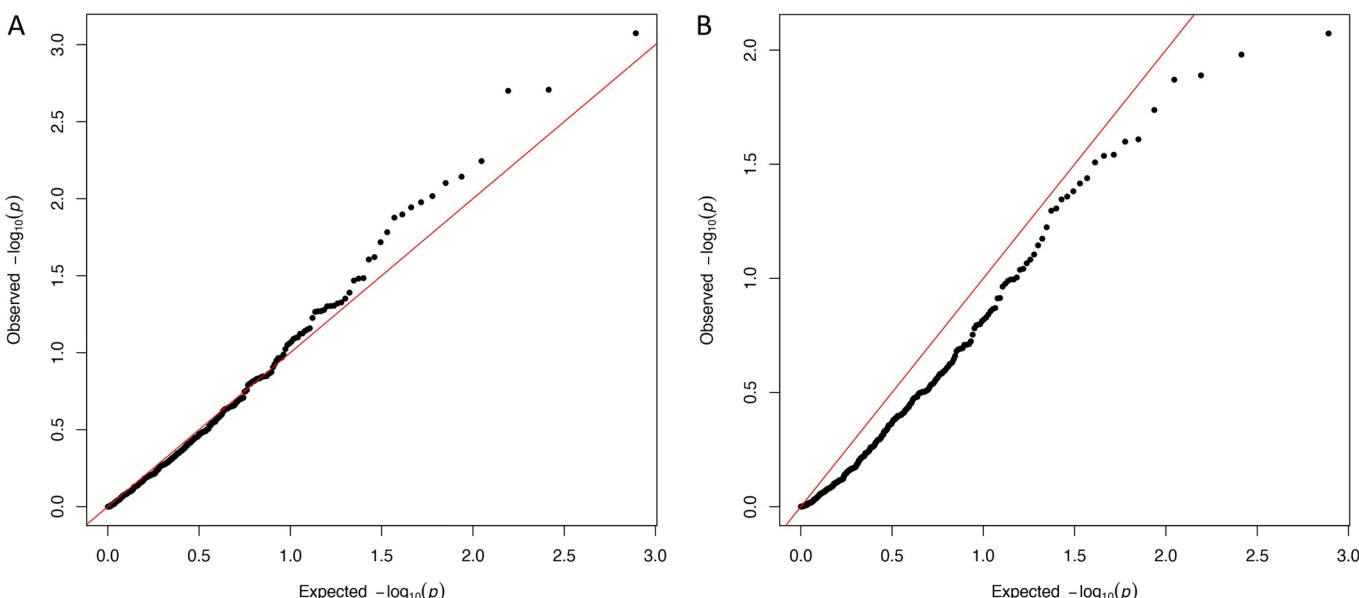

**Extended Data Fig. 3 | QQ plot showing the ultra-rare synonymous burden test among 404 selected genes between SPARK cases and gnomAD controls for allele frequency<1e-5.** *HMCN2* is excluded (not shown) since it has poor coverage in gnomAD. Panel A shows the cross-ancestry case-control ultra-rare synonymous burden comparison, while Panel B shows the European-only case-control ultra-rare synonymous burden comparison. The observed P values for each gene are sorted from largest to smallest and plotted against expected values from a theoretical chi-square distribution.

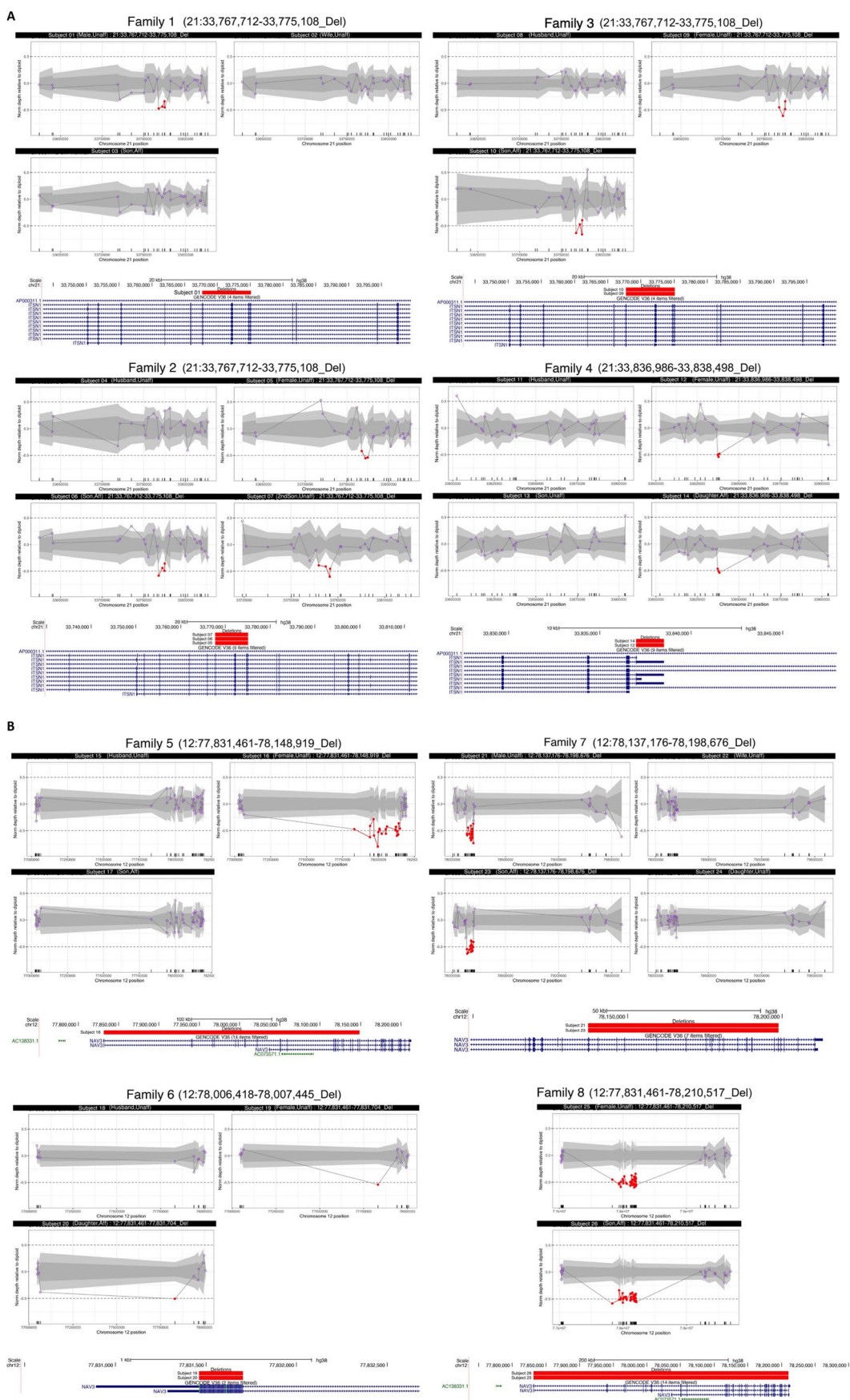

**Extended Data Fig. 4 | See next page for caption.**

**Extended Data Fig. 4 | Transmission disequilibrium of exonic or single gene deletions of ITSN1 (A) and NAV3 (B).** The read depth signal plots show normalized read depth (NDP) of exome targets used in CNV calling by CLAMMS[84] for *ITSN1* (A) and *NAV3* (B) in Family 1–8. NDPs of −0.5, 0 and 0.5 correspond to copy number (CN) of 1 (deletion), 2 (normal diploid) and 3 (duplication). NDPs of exon targets within deleted regions are colored red. Dark and gray areas correspond to 1 and 2 estimated standard deviations of NDP for each exon target. CN deletions were initially discovered from all unaffected parents, then subsequently genotyped on Family 1–8. Signal plots of Family 1–8 are shown with parents appear in the top and offspring in the bottom. The deletion regions and affected exons of genes are shown below each plot.

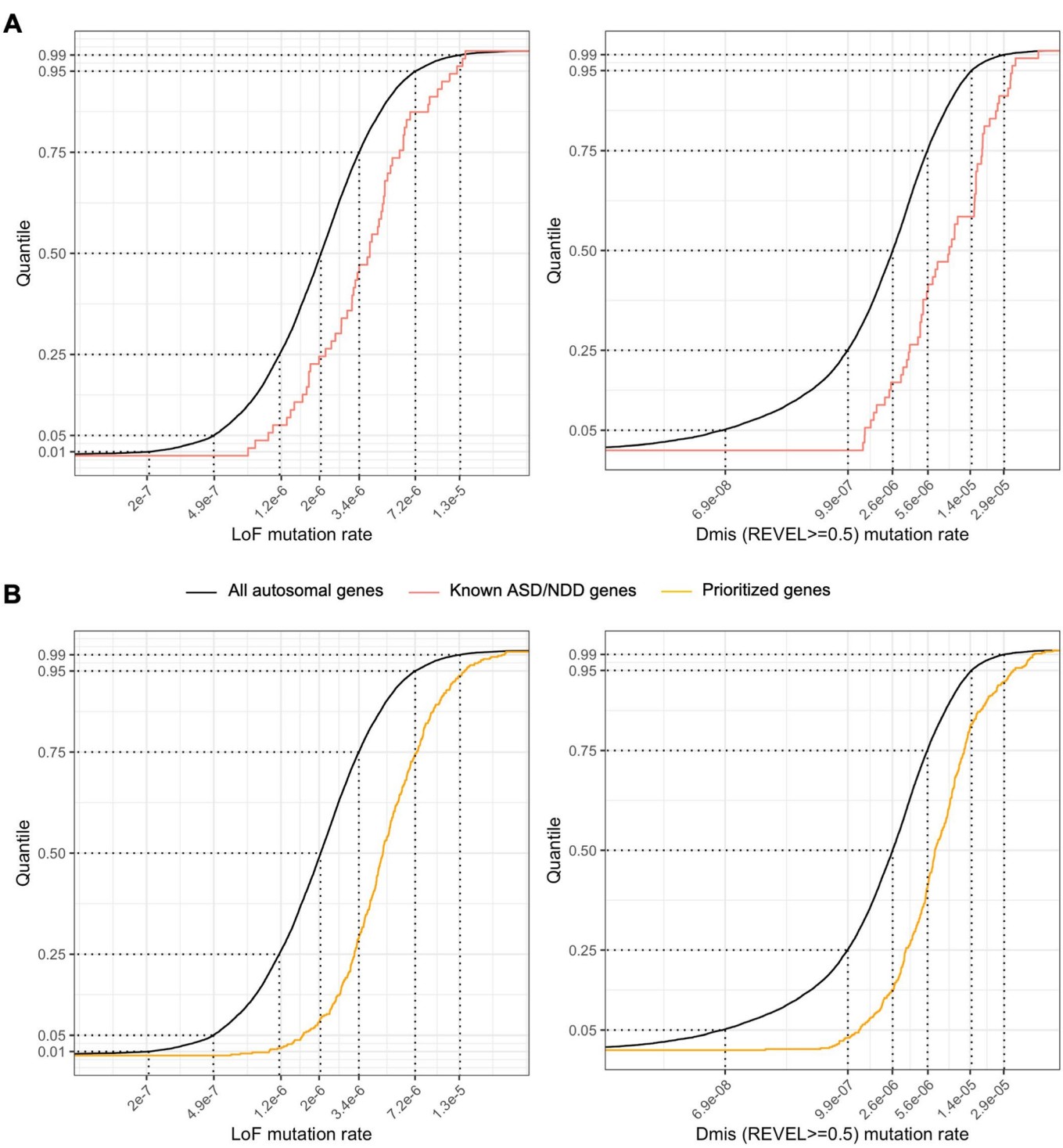

**Extended Data Fig. 5 | Cumulative distribution of haploid mutation rates (per generation) of LoF and D-mis (REVEL > = 0.5) variants of all protein coding genes on autosomes.** Panel A shows the distribution of all autosomal genes vs. known ASD/NDD genes. Panel B shows all autosomal genes vs. prioritized genes. Baseline mutation rates were calculated using 7mer sequence context dependent mutation rates[55]. Known ASD/NDD genes tend to have higher mutation rates than average genes.

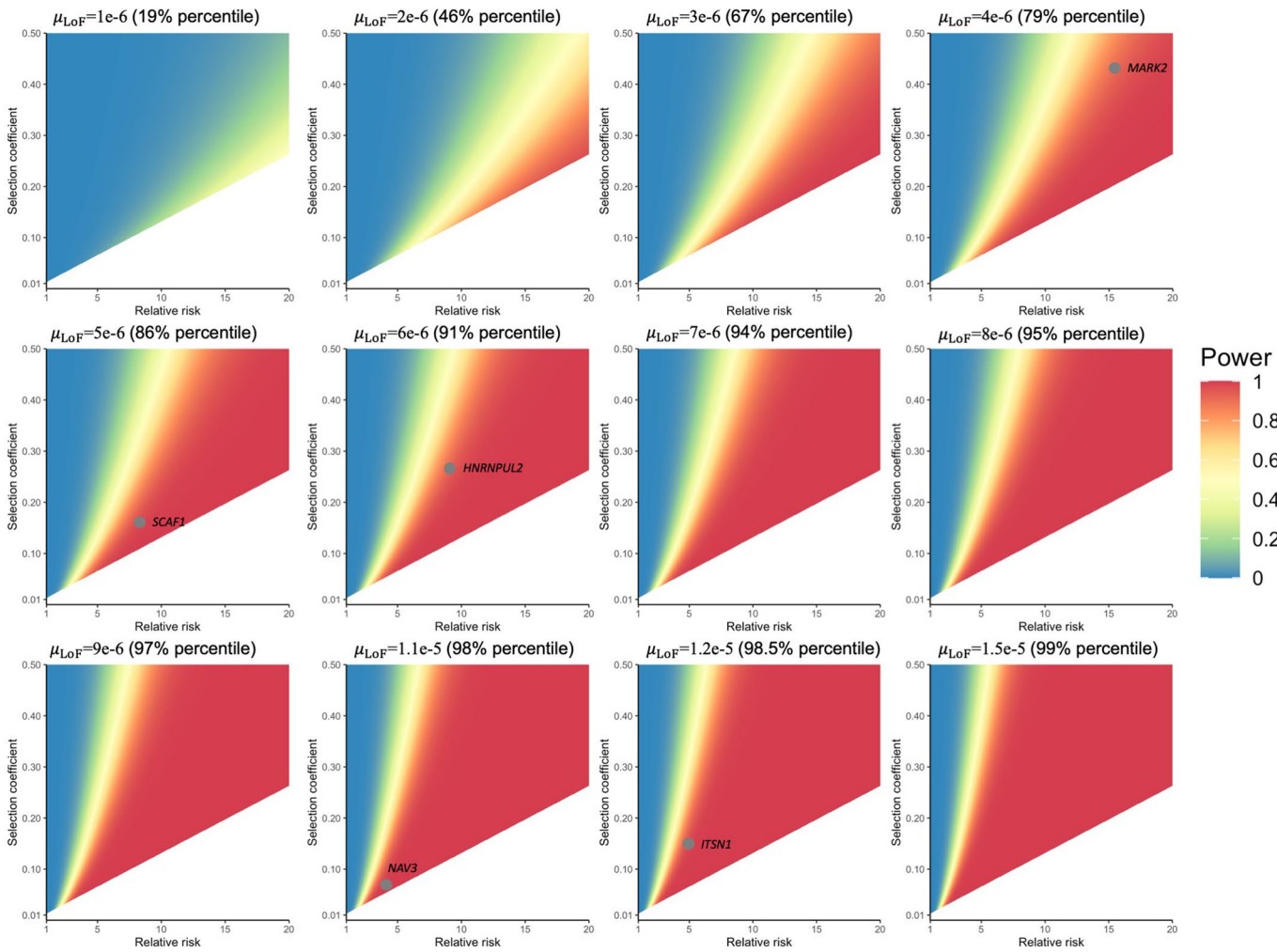

**Extended Data Fig. 6 | Power of case-control association by rare LoFs variants ('mega-analysis') with sample size equal to current study.** The mega-analysis of current study compares the rate of LoF variants in 32,024 unrelated ASD cases with population controls with sample sizes about 76,000~132,000. For power calculation, we assumed that population controls are infinite so that cumulative allele frequency are known and presumed to be at equilibrium under selection-mutation balance for constrained genes ($f = \mu_{LoF}/s$). Experiment-wide error rate was set at 9e-6 (0.05 divided by the number of autosomal genes at gnomAD LOEUF 30%). Power is calculated as a function of relative risk for ASD (*RR*) and selection coefficient (*s*) across different haploid LoF mutation rates ($\mu_{LoF}$) using an analytic approximation by Zuk et al.[41]. We only considered selection coefficient between 0.01 and 0.5 and relative risk to ASD between 1 and 20, because genes with huge effect sizes and larger selection coefficients are expected to be identified from the enrichment of de novo variants. The triangular region where $s < 0.013RR$ are left blank because the parameters in this region are not compatible with the current estimates of prevalence of ASD (1/54)[71] and sex-averaged reduction of reproductive fitness (0.71)[72]. Five new ASD genes identified in this study are placed onto the heatmap closest to its LoF mutation rate. Their positions within heatmaps are taken from point estimates of using gnomAD exomes (non-neuro subset) as population controls.

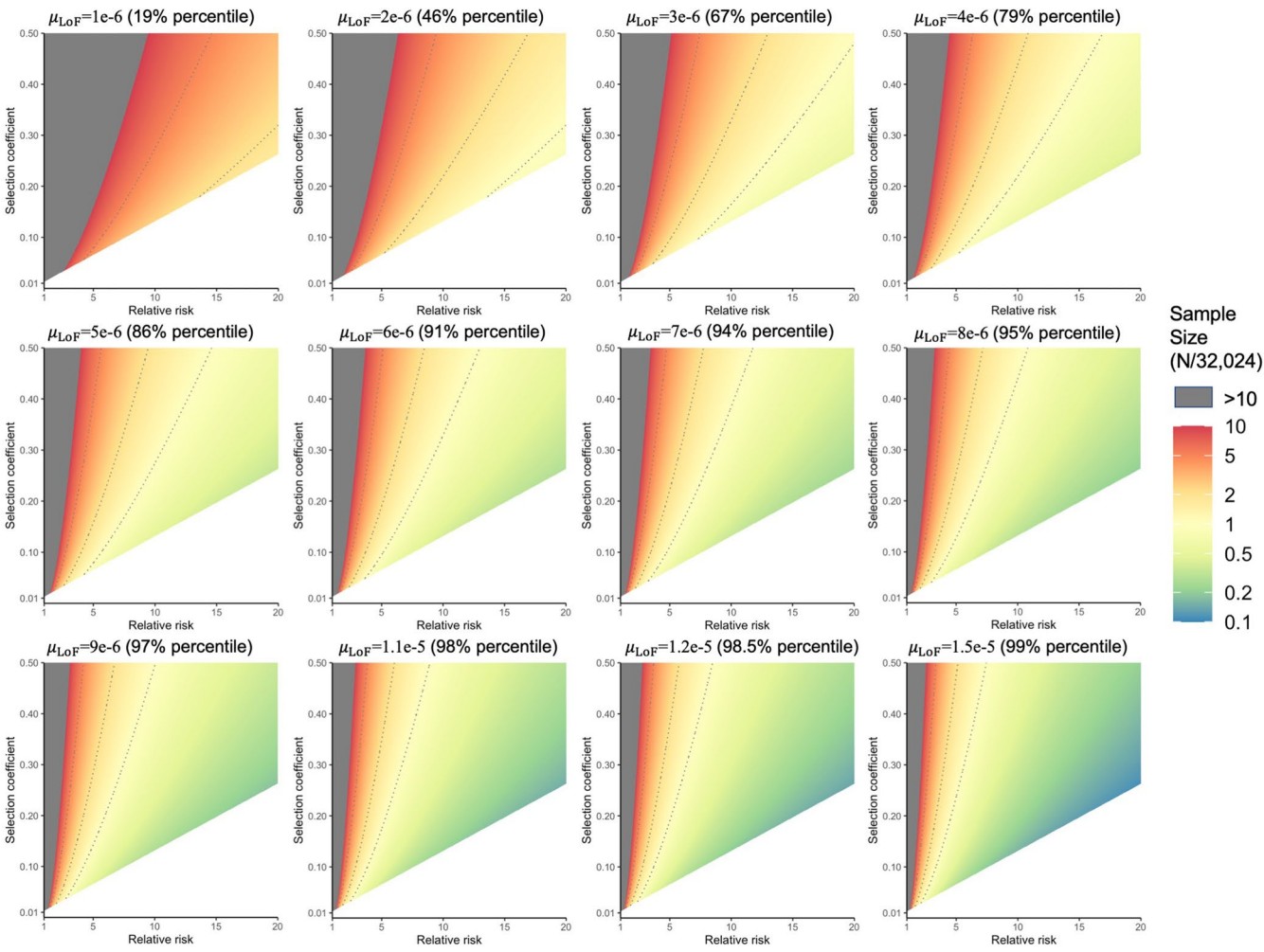

**Extended Data Fig. 7 | Sample sizes required for achieving 90% of power.** Using the same assumptions and experiment-wide error rate as power calculation, we calculated required sample size for 90% of power. Sample size is shown as a factor relative to the current sample size (32,024) and as a function of relative risk to ASD and selection coefficient across with different LoF mutation rates. Contours 1, 2 and 5 times of current size are shown as dashed lines. Regions of parameter space that require over 10 times current sample sizes are shown in gray.

# Reporting Summary

## Statistics

For all statistical analyses, confirm that the following items are present in the figure legend, table legend, main text, or Methods section.

| n/a | Confirmed | |
|---|---|---|
| ☐ | ☒ | The exact sample size ($n$) for each experimental group/condition, given as a discrete number and unit of measurement |
| ☒ | ☐ | A statement on whether measurements were taken from distinct samples or whether the same sample was measured repeatedly |
| ☐ | ☒ | The statistical test(s) used AND whether they are one- or two-sided *Only common tests should be described solely by name; describe more complex techniques in the Methods section.* |
| ☐ | ☒ | A description of all covariates tested |
| ☐ | ☒ | A description of any assumptions or corrections, such as tests of normality and adjustment for multiple comparisons |
| ☐ | ☒ | A full description of the statistical parameters including central tendency (e.g. means) or other basic estimates (e.g. regression coefficient) AND variation (e.g. standard deviation) or associated estimates of uncertainty (e.g. confidence intervals) |
| ☐ | ☒ | For null hypothesis testing, the test statistic (e.g. $F$, $t$, $r$) with confidence intervals, effect sizes, degrees of freedom and $P$ value noted *Give P values as exact values whenever suitable.* |
| ☐ | ☒ | For Bayesian analysis, information on the choice of priors and Markov chain Monte Carlo settings |
| ☐ | ☒ | For hierarchical and complex designs, identification of the appropriate level for tests and full reporting of outcomes |
| ☐ | ☒ | Estimates of effect sizes (e.g. Cohen's $d$, Pearson's $r$), indicating how they were calculated |

*Our web collection on statistics for biologists contains articles on many of the points above.*

## Software and code

Policy information about availability of computer code

| Data collection | All participants were recruited to SPARK under a centralized IRB protocol (Western IRB Protocol #20151664). All participants provided written informed consent to take part in the study. Written informed consent was obtained from all legal guardians or parents for all participants age 18 and younger and all participants age 18 and older who have a legal guardian. Assent was also obtained from dependent participants age 10 and older. Participants with autism were compensated $25-50 depending on other registered family members. Saliva samples were collected from participants and used for DNA extraction. New exome sequencing samples in this study were captured by IDT xGEN research panel and sequenced on Illumina NovaSeq. DNA samples were also genotyped for over 600K SNPs by Infinium Global Screening Array. |
|---|---|
| Data analysis | All software used in this study is publicly available. Supplementary Table S16 describes software versions and parameter settings used. The code for major figures and analysis under https://github.com/ShenLab/SPARK_Analysis_V1.git. The DOI is 10.5281/zenodo.6646871. |

For manuscripts utilizing custom algorithms or software that are central to the research but not yet described in published literature, software must be made available to editors and reviewers. We strongly encourage code deposition in a community repository (e.g. GitHub). See the Nature Portfolio guidelines for submitting code & software for further information.

## Data

Policy information about availability of data

All manuscripts must include a data availability statement. This statement should provide the following information, where applicable:
- Accession codes, unique identifiers, or web links for publicly available datasets
- A description of any restrictions on data availability
- For clinical datasets or third party data, please ensure that the statement adheres to our policy

In order to abide by the informed consents that individuals with autism and their family members signed when agreeing to participate in a SFARI cohort (SSC and

SPARK), researchers must be approved by SFARIbase (https://base.sfari.org). To access to SPARK/SFARI data, researchers should
1) Obtain a SFARI base account at https://base.sfari.org, which will require affiliating with an institution. Currently there are 271 institutions around the world that have signed SFARI's RDA and any researcher affiliated with those institutions can apply for SFARI base access.
2) Review the institute's executed Researcher Distribution Agreement (RDA). The standard RDA is here: https://s3.amazonaws.com/sf-web-assets-prod/wp-content/uploads/sites/2/2021/06/15165956/SFARI_RDA.pdf
3) Create a SFARI base project, which includes a title, abstract and an IRB approval or exemption document.
4) Create a SFARI base request. All requests are processed in a timely manner.

The SPARK data is accessible as follows:
SFARI_SPARK_iWES: This includes exome and genotyping data on 70,487 participants, including all people analyzed in this paper plus an additional 11,282 participants.
SFARI_SPARK_WGS_1: This includes whole genome data from 2,629 individuals from 645 families with at least one person with autism.
SFARI_SPARK_WGS_2: This includes whole genome data from 2,365 individuals, from 587 families with at least one person with autism.
SFARI_SPARK_WGS_3: This includes whole genome data from 2,871 individuals, from 803 families with at least one person with autism.
SSC_WES_3: This is whole exome data on the Simons Simplex Collection (SSC) as analyzed and reported by Krumm et al, 2015.
SFARI_SSC_WGS_pilot: This is genomes of 40 families of autism.
SFARI_SSC_WGS_1 and SFARI_SSC_ WGS_2: WGS of the SSC.
SSC Dataset: Phenotypic information on 2,644 simplex autism families.
SPARK Phenotype Dataset V7: This is the current phenotypic dataset on 290,502 SPARK participants, including 111,720 participants with autism.

# Field-specific reporting

Please select the one below that is the best fit for your research. If you are not sure, read the appropriate sections before making your selection.

☒ Life sciences    ☐ Behavioural & social sciences    ☐ Ecological, evolutionary & environmental sciences

For a reference copy of the document with all sections, see nature.com/documents/nr-reporting-summary-flat.pdf

# Life sciences study design

All studies must disclose on these points even when the disclosure is negative.

| | |
|---|---|
| Sample size | The sample size of people with autism is the largest genomic analysis of people with autism to date. The SPARK cohort represents the participants that registered, provided informed consent and provided a DNA sample between 2015-2019. |
| Data exclusions | We excluded parents with autism and intellectual disability from the TDT analysis. |
| Replication | A set of 400 genes was prioritized in the first stage of the analysis. These genes were analyzed in the 2nd stage of analysis. The experiments have not been replicated although other papers (Fu et al, in press at Nature Genetics) have performed complementary analyses with consistent results. |
| Randomization | All SPARK participants reside in the US, speak English and reported whether or not they had a professional diagnosis of autism. Covariates are not relevant to the genetic basis for autism as the group includes a very large number of diverse participants. |
| Blinding | Participants registered in the study as having autism or not so we were unable to blind the study. All families in SPARK had to have at least one family member with a professional diagnosis of autism. |

# Reporting for specific materials, systems and methods

We require information from authors about some types of materials, experimental systems and methods used in many studies. Here, indicate whether each material, system or method listed is relevant to your study. If you are not sure if a list item applies to your research, read the appropriate section before selecting a response.

### Materials & experimental systems

| n/a | Involved in the study |
|---|---|
| ☒ | Antibodies |
| ☒ | Eukaryotic cell lines |
| ☒ | Palaeontology and archaeology |
| ☒ | Animals and other organisms |
| ☐ | ☒ Human research participants |
| ☒ | Clinical data |
| ☒ | Dual use research of concern |

### Methods

| n/a | Involved in the study |
|---|---|
| ☒ | ChIP-seq |
| ☒ | Flow cytometry |
| ☒ | MRI-based neuroimaging |

# Human research participants

| | |
|---|---|
| Population characteristics | Any person with a professional diagnosis of autism residing in the US and speaking English was eligible for this study. The mean age of the SPARK cohort in this analysis was 16.5 years old (SD 19.2 years). Cases made up 46% of the SPARK cohort (the remaining 54% were controls). The sex breakdown of the full SPARK cohort was 57% male and 43% female. The sex breakdown of the SPARK case cohort was 77% male and 23% female. |
| Recruitment | All participants were recruited by clinical sites receiving a grant from SFARI or through digital and social media or another advertisement. All recruitment materials were approved by Western IRB (http://wirb.com/) under Protocol ##20151664). All participants provided written informed consent to take part in the study. Written informed consent was obtained from all legal guardians or parents for all participants age 18 and younger and all participants age 18 and older who have a legal guardian. Assent was also obtained from dependent participants age 10 and older.  The individuals with autism and/or their parents/guardians/siblings that participate in SPARK may be more interested in scientific research than other people. This should not impact the findings of the study. |
| Ethics oversight | All participating institutions waived IRB oversight to a central IRB, which was Western IRB, (http://wirb.com/). |

Note that full information on the approval of the study protocol must also be provided in the manuscript.

