## [Peer Review File · Nature Genetics]

Peer Review Information

Manuscript Title: Integrating de novo and inherited variants in over 42,607 autism cases identifies mutations in new moderate risk genes

Corresponding author name(s): Dr Wendy Chung

Reviewer Comments & Decisions:

Decision Letter, initial version:

9th Dec 2021

Dear Dr Feliciano,

First of all, I am so sorry that it has taken so long to return this decision to you. Thank you so much for your patience.

Your Article, "Integrating de novo and inherited variants in over 42,607 autism cases identifies mutations in new moderate risk genes" has now been seen by 2 referees. You will see from their comments below that while they find your work of interest, some important points are raised. We are interested in the possibility of publishing your study in Nature Genetics, but would like to consider your response to these concerns in the form of a revised manuscript before we make a final decision on publication.

Overall, you'll see that both reviewers are supportive of the paper and both recognise that it is likely to have real impact in the field. Neither have flagged major technical problems, but both have indicated areas where further analysis or expanded discussion will fortify the conclusions and improve clarity. We would expect all their comments to be addressed in full either experimentally or textually (where appropriate).

We therefore invite you to revise your manuscript taking into account all reviewer and editor comments. Please highlight all changes in the manuscript text file. At this stage we will need you to upload a copy of the manuscript in MS Word .docx or similar editable format.

*2) If you have not done so already please begin to revise your manuscript so that it conforms to our Article format instructions, available [here](http://www.nature.com/ng/authors/article_types/index.html). Refer also to any guidelines provided in this letter.

[REDACTED]

We look forward to seeing the revised manuscript and thank you for the opportunity to review your work. Again, I am so sorry that this decision took so long to reach you.

Sincerely,

Safia Danovi

2Editor
Nature Genetics

Referee expertise:

Referee #1: Autism genetics

Referee #2: statistical genetics

Reviewers' Comments:

Reviewer #1:
Remarks to the Author:

The paper reports on the largest sequencing study of ASD to date, including 42,607 cases, of whom 35,130 are new cases recruited by SPARK. The authors analyze rare de novo and inherited coding variants in a two-stage design and identify 60 exome-wide significant genes, including five new risk genes. The majority (four) of the five new risk genes confer moderate risk by rare deleterious variants. Their association is driven in one case (the NAV3 gene) entirely by inherited loss-of-function (LoF) variants, in another case (MARK2) primarily by de novo variants (DNVs), while the remaining three are supported by both DNVs and rare LOFs. The authors find that known ASD or neurodevelopmental disorder risk genes explain around 70% of the genetic burden conferred by DNVs, while < 20% of the genetic risk conferred by rare inherited LoF variants is located in these known risk genes. Based on these results, the authors conduct power calculations estimating the large number of cases (and controls) needed to identify additional genes that confer moderate risk by rare and predominantly inherited variants.

This is a comprehensive, well-conducted and thorough study with interesting analyses and important results that undoubtedly will be of interest to the field.

However, I have some concern regarding potential population substructure bias in the case-control analysis:

- The thorough assessment of substructures, variant rates and characteristics in the two reference populations is highly appreciated. However, I still have some concern regarding the case-control analysis, particularly regarding potential bias from ancestral substructures in the case group. For instance, if some rare LoFs are wrongly classified as high-confidence and are actually non- or weakly pathogenic/deleterious variants (in spite of the applied filtering), they might be predominantly/exclusively present in specific ancestry groups. Thus, I think it would be relevant to (i) perform the case-control analysis separately in EUR only and in other main ancestries as a sensitivity analysis, (ii) check how the alleles that contribute to the (meta-)analysis of the five novel genes partition on ancestry groups (primarily in cases), and (iii) check how the rare synonymous variant association looks in the case-control analysis (in Europeans only and in the cross-ancestry analysis).
- It would also be informative to see how the PCA of cases and the two control groups compares

3(similar to the Suppl Fig S22).

Other comments:

- The authors find that the population attributable risk (PAR) from damaging DNVs is ~10%. It would be informative to discuss how that compares to the previously reported estimate that DNVs explain ~2% of variance in ASD liability.
- What is the rationale in excluding parents with ASD or ID in the analysis of inherited LoF variants? How many are excluded on those criteria? And why is the MSSNG cohort not part of the analysis?
- The 7th LOEUF decile seems to show a significant under-transmission to affected offspring. How do you explain that? It could perhaps be due to a protective effect of the variants and it would be interesting to see gene-wise association results for those variants...perhaps listing the top-ranking genes and investigating the functional/biological characteristics of that gene set.
- The 10th LOEUF seems to show a significant under-transmission to unaffected offspring. How can that be explained? Technical bias? Random fluctuation?
- The authors state that the association of NAV3 with ASD risk is entirely driven by rare inherited variants and that three other novel genes have contributions from inherited variants. As the contribution from the case-control analysis may arise from both de novo and inherited variants, I think it should be mentioned somewhere that a de novo variant contribution to the NAV3 association (and the other three) cannot be excluded.
- I might have missed it, but what is the specific selection criteria for the nine highly-penetrant genes (in Figure 6A)?
- How well does the method used for identifying sample overlap in the de novo analysis work? How many overlapping individuals were identified?

Minor:

- The title says "...over 42,607 autism cases...". It appears to be the exact number of cases included, so I'd suggest to delete "over".
- It seems from the Results text that Figure 3 should be renamed Figure 4 and vice versa.
- There seems to be some discrepancy in the reference list. "...(STRING102) and phenotypic (HPO103)..." should cite #100 and 102 in the reference list.
- In Discussion it is stated that "...the data show that variants in these new ASD genes have effects on core symptoms of ASD, cognition, and other behaviors including schizophrenia, Tourette syndrome, ADHD and other behavioral conditions."

I cannot seem to locate the results that substantiate the mentioning of ADHD and other behavioral

4conditions..? Also, I'd suggest to replace "behaviors" with "mental disorders" (or similar) in the sentence "...ASD, cognition, and other behaviors including..."

- I think the sentence in Methods explaining pExt ("pExt for each variant can be operationally defined as the proportion of expression levels of transcripts whose variant effects are the same as gene effect over all transcripts included in the annotation²⁷") could be clearer.

Reviewer #2:

Remarks to the Author:

The manuscript by Chung, Shen and colleagues presented the largest ASD gene discovery to date. It is a rigorous study with interesting findings, particularly about inherited LoF variants. Notably, it reported that while known ASD/DDD genes explained the majority of DNV burden, they explain only 20% of burden in inherited LOFs, suggesting many missing ASD genes. It also shows that it is possible to discover ASD genes using rare LoF variants from families or case-control studies, a finding that has implications on future study design. Overall, the interesting findings as well as the resources provided in this paper would make it appealing to the audience of Nature Genetics.

Major comments:

- 1) The main analysis is done in two stages, with the first stage identifying about 400 genes, and then in the second stage, meta-analysis is done on these genes to combine evidence from multiple tests. I don't see the benefit of this two-stage analysis, however. I would imagine that the smaller gene set from the first stage may reduce multiple testing burden in the second stage, potentially increasing the power of study. But in fact, in the second stage of meta-analysis, the authors still apply exome-wide significance threshold, so it doesn't really benefit from gene selection in the first stage. It seems straightforward to just apply the meta-analysis on all genes. This needs to be better explained.
- 2) The authors used a simple binomial test for LoF variants in case-control data. The concern here is that the cases and controls are from two different studies with potentially different biases and rates of calling LoF variants. This may lead to inflated p-values. The authors said that AFs of the LoF variants have similar distributions in cases and controls. This is certainly helpful, still, I think it would be more rigorous to show directly that the results are not inflated. A QQ plot of p-values from all genes would be helpful.
- 3) In the meta-analysis, the three tests are combined using Fisher's method. While this is understandable, given that standard fixed effect meta-analysis is not applicable, Fisher's method is not optimal, especially when combining tests with different power. This may happen for example, for the genes that are driven by LoF DNVs, but contain very few inherited LoFs, thus the TDD/case-control test would have low power. It would be useful to assess how much is gained from meta-analysis. For example, if one just uses DNV data, how many genes are found? Similarly, how many are found using only inherited LoF variants (combining TDT and case-control)? How many genes are found using a single test, but actually missing in meta-analysis?

54) The archetypal analysis is interesting. It nicely projects the functions of genes into 6 archetypes. However, the method is not explained clearly. This is especially a concern, given that this is not a common analysis (at least in the human genetics literature). Some specific comments and suggestions include: it would be useful to give a high-level description of the archetypal analysis to the readers (as a general background). In Table S10, there are 500 clusters. I suppose these clusters are found by STRING, but 500 seems a very large number of clusters. How are these clusters defined? What are the genes in each cluster? Also some rows of table S10 contain a single cluster, while others have more, why? How did the authors obtain "description" of each row?

5) Another concern about the archetypal analysis: the authors have simple interpretations of each archetype, e.g. A6 is for KRAB domain and LRR. This makes it easy to follow the functions of genes. However, I feel that this may be an over-simplification. From Table S10: some archetypes seem to be quite functionally heterogeneous. For example, the authors annotate A4 as "membrane trafficking and protein transport", but the clusters associated with A4 have many other functions, particularly a number of signaling pathways such as Wnt, JAK-STAT, TGF-beta, interferon and BMP signaling. A6 seems to be even more heterogeneous, with functions including Olfactory signaling, homeodomain, Ca²⁺ binding, cell polarity, glycosylation and so on. So annotating it as "KRAB domain and LRR" seems a bit misleading here.

Minor comments

Fig. 3 and 4 are in the wrong order.

Fig. 4: right panel, "percentage of over-transmission of rare LoFs to cases". Why the percents there are sometimes so high, reaching 50-60%? How is "over-transmission" defined? If I understand correctly, it should be percent of transmitted vs. non-transmitted, but then 50-60% seems too high. Fig. 2B, for examples, show much smaller over-transmission.

Fig. 5 legend: in my Word file, I see, "All five novel genes (Error! Reference source not found)".

Reviewer #3:
None

Author Rebuttal to Initial comments

Reviewer #1:

Remarks to the Author:

The paper reports on the largest sequencing study of ASD to date, including 42,607 cases, of whom 35,130 are new cases recruited by SPARK. The authors analyze rare de novo and

6inherited coding variants in a two-stage design and identify 60 exome-wide significant genes, including five new risk genes. The majority (four) of the five new risk genes confer moderate risk by rare deleterious variants. Their association is driven in one case (the NAV3 gene) entirely by inherited loss-of-function (LoF) variants, in another case (MARK2) primarily by de novo variants (DNVs), while the remaining three are supported by both DNVs and rare LOFs. The authors find that known ASD or neurodevelopmental disorder risk genes explain around 70% of the genetic burden conferred by DNVs, while < 20% of the genetic risk conferred by rare inherited LoF variants is located in these known risk genes. Based on these results, the authors conduct power calculations estimating the large number of cases (and controls) needed to identify additional genes that confer moderate risk by rare and predominantly inherited variants.

This is a comprehensive, well-conducted and thorough study with interesting analyses and important results that undoubtedly will be of interest to the field.

However, I have some concern regarding potential population substructure bias in the case-control analysis:

- The thorough assessment of substructures, variant rates and characteristics in the two reference populations is highly appreciated. However, I still have some concern regarding the case-control analysis, particularly regarding potential bias from ancestral substructures in the case group. For instance, if some rare LoFs are wrongly classified as high-confidence and are actually non- or weakly pathogenic/deleterious variants (in spite of the applied filtering), they might be predominantly/exclusively present in specific ancestry groups. Thus, I think it would be relevant to (i) perform the case-control analysis separately in EUR only and in other main ancestries as a sensitivity analysis, (ii) check how the alleles that contribute to the (meta-)analysis of the five novel genes partition on ancestry groups (primarily in cases), and (iii) check how the rare synonymous variant association looks in the case-control analysis (in Europeans only and in the cross-ancestry analysis).

Response: Thank you for your thoughtful comments. Yes, we agree that population structure and technical batch effect are important issues in case-control association analyses. We have performed sensitivity analyses in European Americans ($N_{\text{case}}=11,027$), African Americans ($N_{\text{case}}=1,027$), and Admixed Americans (Latinx) ($N_{\text{case}}=2,423$). The sample sizes of the East Asian Americans and South Asian Americans are too small for meaningful effect size estimation.

For cases, we inferred ancestry using principal component analysis as described in the manuscript. For population controls (gnomAD 2.1.1, non-neuro exome), we used ancestry

provided by the gnomAD website (**Supplementary Table S10**). We performed association tests in each ancestry separately, and then combined p-values using Fisher's method. Of the five novel genes, the estimated effect sizes within cases of European ancestry are close to the overall sizes reported in our previous submission (**Supplementary Table S10**). The combined p-values are similar to those observed in the overall analysis as well, especially for *NAV3*, *SCAF1*, and *ITSN1*. We added the following line on page 7 in the manuscript to describe the ancestry-specific analyses: "The combined p-values based on ancestry-specific case-control analyses are similar to the overall case-control analysis for these five genes (**Supplementary Table S10**)."

We further compared the burden of ultra-rare synonymous variants (allele frequency < 1e-5) between ASD cases and gnomAD controls in the 404 selected genes (**Supplementary Figure S11**). Most genes showed similar burden between cases and controls except for *HMCN2* (not shown) which has poor coverage in gnomAD. The overall ultra-rare synonymous rate is 0.57 for both ASD cases and gnomAD controls for all ancestry groups. The burden of ultra-rare synonymous variants is also similar between European ancestry cases and controls. We added the following line on page 6 in the manuscript to describe the ultra-rare synonymous burden: "The ultra-rare synonymous variant burden is similar between cases and controls across the selected genes (**Supplementary Figure S11**)."

Overall, the case-control association results of ultra-rare LoF variants are robust against potential population structure and technical batch effects.

Table S10. Ancestry specific case-control analysis for five novel genes.

Gene	All Ancestry				European Ancestry				African Ancestry			American Admixed			Combined p value for each Ancestry-specific analysis
	Case Rate (%)	Control Rate (%)	Relative Risk	p value	Case Rate (%)	Control Rate (%)	Relative Risk	p value	Case Rate (%)	Control Rate (%)	p value	Case Rate (%)	Control Rate (%)	p value	
NAV3	1.39	0.30	4.6	4.39E-07	1.36	0.33	4.1	2.44E-04	0.00	0.12	1.00E+00	2.48	0.39	2.90E-03	2.95E-07
MARK2	0.25	0.02	12.5	4.49E-03	0.27	0.04	6.8	6.00E-02	0.00	0.00	1.00E+00	0.41	0.00	1.37E-01	2.20E-02
SCAF1	0.70	0.03	23.3	2.07E-06	0.91	0.02	45.5	1.84E-05	1.95	0.00	1.26E-02	0.00	0.00	1.00E+00	4.47E-06
ITSN1	0.63	0.16	3.9	1.99E-03	0.54	0.20	2.7	7.00E-02	0.00	0.62	1.00E+00	1.65	0.00	3.52E-04	8.10E-05
HNRNPUL2	0.63	0.04	15.8	2.58E-06	0.82	0.07	11.7	1.26E-04	0.00	0.00	1.00E+00	0.00	0.00	1.00E+00	2.49E-04

Supplementary Figure S11: QQ plot showing the ultra-rare synonymous burden test between SPARK cases and gnomAD controls. Panel A: cross-ancestry case-control ultra-rare synonymous burden comparison; Panel B: European American-only case-control ultra-rare synonymous burden comparison.

- It would also be informative to see how the PCA of cases and the two control groups compares (similar to the Suppl Fig S22).

Response: Unfortunately, we don't have access to the individual level data among the control groups, and we are not able to extract PCAs from controls. While this prevents a perfect match in genetic ancestry between cases and controls, the results from burden analysis of rare synonymous variants support that the impact of ancestry to association is minimal.

Other comments:

- The authors find that the population attributable risk (PAR) from damaging DNVs is ~10%. It would be informative to discuss how that compares to the previously reported estimate that DNVs explain ~2% of variance in ASD liability.

Response: Thanks for pointing this out as we think we can better clarify the differences between the two numbers to which you are referring. There is a conceptual difference in PAR from damaging DNVs (de novo variants) and variance explained by DNV in ASD liability, and these two values are calculated quite differently (please see the following references: 1) Rockhill, B., Newman, B., & Weinberg, C. (1998). Use and misuse of population attributable fractions. *American journal of public health*, 88(1), 15-19.; 2) Dahlqvist, E., Magnusson, P. K., Pawitan, Y., & Sjölander, A. (2019). On the relationship between the heritability and the attributable fraction. *Human genetics*, 138(4), 425-435.).

PAR is defined as the proportion of the incidence of ASD in the population that is due to damaging DNVs. In our study, PAR is approximated by the difference of the frequency of damaging DNVs among ASD cases and controls. In contrast, variance explained by DNVs is defined based on the liability threshold model with the assumption that each individual has an underlying liability to acquire ASD, and the variance explained by DNVs is the estimated contribution of DNVs to the overall ASD risk. Given the effect size of a class of variants, PAR is proportional to the cumulative frequency of the variants, whereas variance explained is quadratic to the frequency. Therefore, it is expected that PAR from DNVs is different from the variance that DNVs explain in ASD liability.

- What is the rationale in excluding parents with ASD or ID in the analysis of inherited LoF variants? How many are excluded on those criteria? And why is the MSSNG cohort not part of the analysis?

Response:

Our rationale for excluding parents with ASD and ID is that the overall number of parents with these conditions is too small to be informative and to increase the power of the transmission equilibrium test. The number of parents with ASD or ID in SPARK is relatively small (N=87, 1.9% of all parents). The MSSNG cohort is not included because we do not have access to the phenotypic data to assess whether or not the parents have intellectual disability.

- The 7th LOEUF decile seems to show a significant under-transmission to affected offspring. How do you explain that? It could perhaps be due to a protective effect of the variants and it would be interesting to see gene-wise association results for those variants...perhaps listing the top-ranking genes and investigating the functional/biological characteristics of that gene set.

Response: Thanks for noticing this in **Figure 2**. We have calculated the p-values for under-transmission among the 7th LOEUF decile to affected offspring, and it is not significant ($p=0.076$). To avoid confusion, we have now marked any statistically significant two-sided transmission disequilibrium test with a star sign in **Figure 2**.

- The 10th LOEUF seems to show a significant under-transmission to unaffected offspring. How can that be explained? Technical bias? Random fluctuation?

Response: Similar to the above response, we calculated the p-value for under-transmission within the 10th LOEUF decile to unaffected offspring, and it is not significant ($p=0.093$). To avoid confusion, we marked any statistically significant two-sided transmission disequilibrium test with a star sign in **Figure 2**.

• The authors state that the association of NAV3 with ASD risk is entirely driven by rare inherited variants and that three other novel genes have contributions from inherited variants. As the contribution from the case-control analysis may arise from both de novo and inherited variants, I think it should be mentioned somewhere that a de novo variant contribution to the NAV3 association (and the other three) cannot be excluded.

Response: We agree and have changed the wording from “entirely” to “primarily” to accurately describe the contribution of inherited variants to NAV3.

• I might have missed it, but what is the specific selection criteria for the nine highly-penetrant genes (in Figure 6A)?

Response: The nine genes selected in **Figure 6A** included six genes that are well-known to harbor *de novo* risk for ASD (*CHD8*, *SHANK3*, *SCN2A*, *ADNP*, *ARID1B* and *FOXP1*) and were the most frequently identified in SPARK, which maximizes the number of samples available for genotype-phenotype analyses. The other 3 genes (*KDM5B*, *GIGYF1* and *KMT2C*) were chosen because they exceed exome-wide significance in the total DeNovoWEST analysis, but also have had previous evidence of inherited risk and significant values in our TDT analysis and are frequently identified in SPARK (**Figure 5A**). We have clarified the text of the manuscript to indicate these criteria for “known *de novo*” and “known inherited” risk genes and included relevant citations in the legend for **Figure 6** by inserting the following sentences:

“The nine known ASD genes include 6 genes (colored pink and labeled *de novo*, known) that are well-established *de novo* ASD risk genes that were most frequently identified in SPARK, which maximizes the number of samples available for genotype-phenotype analyses. We also included 3 genes (colored light blue and labeled inherited, known) that have some previous evidence for inherited ASD risk (*GIGYF1*¹⁷, *KDM5B*⁵⁷ and *KMT2C*⁵⁸) and were also frequently identified in SPARK.”

• How well does the method used for identifying sample overlap in the de novo analysis work? How many overlapping individuals were identified?

Response: We identified the sample overlap by two methods: (a) when exome, SNP array, or genome data are available, we inferred relatedness by genotypes and exclude apparent duplicates within and across cohorts. This method is applicable to SPARK and SSC cohorts. (b) when genome-wide data are not available (ASC and MSSNG cohorts), we identify sample overlap based on shared *de novo* variants between cohorts. This method would eliminate double counting of the same *de novo* variants from the same individual and is conservative to avoid any inflation in our *de novo* analysis due to unaccounted sample overlap. By this method,

11

we identified 23 samples overlapping with shared *de novo* mutation between SPARK and MSSNG, 17 samples overlapping between SPARK and ASC, 1 sample overlapping between SSC and MSSNG, 3 samples overlapping between SSC and ASC, and 233 samples overlapping between ASC and MSSNG.

Minor:

- The title says "...over 42,607 autism cases...". It appears to be the exact number of cases included, so I'd suggest to delete "over".

Response: Agreed, we have removed "over" from the title.

- It seems from the Results text that Figure 3 should be renamed Figure 4 and vice versa.

Response: Thank you for pointing this out; this is now fixed.

- There seems to be some discrepancy in the reference list. "...(STRING102) and phenotypic (HPO103)..." should cite #100 and 102 in the reference list.

Response: Thank you, the references to STRING and HPO terms are now fixed and are references 52 and 53 respectively.

- In Discussion it is stated that "...the data show that variants in these new ASD genes have effects on core symptoms of ASD, cognition, and other behaviors including schizophrenia, Tourette syndrome, ADHD and other behavioral conditions."

I cannot seem to locate the results that substantiate the mentioning of ADHD and other behavioral conditions..? Also, I'd suggest to replace "behaviors" with "mental disorders" (or similar) in the sentence "...ASD, cognition, and other behaviors including..."

Response: Thank you for pointing this out. We have changed the language to clarify the point that although the new "rare, inherited" ASD genes have a lower effect size, many individuals with these variants are still affected with additional co-morbid psychiatric conditions in addition to autism. We also point the reader to **Supplementary Table S9** where individual phenotypic information can be found, including epilepsy, schizophrenia, ADHD, cognitive impairment and Tourette syndrome. The sentence now reads, "Nevertheless, the data show that many individuals with variants in these new ASD genes are affected with various neuropsychiatric conditions such as epilepsy, schizophrenia, Tourette syndrome, and ADHD (**Supplementary Table S9**)."

• I think the sentence in Methods explaining pExt (“pExt for each variant can be operationally defined as the proportion of expression levels of transcripts whose variant effects are the same as gene effect over all transcripts included in the annotation²⁷”) could be clearer.

Response: Thank you. We have changed the language and the sentence now reads,

“The pExt (proportion expressed across transcripts) score for each variant is operationally defined as the sum of the expression of all transcripts that include the variant, normalized by the expression of the gene in all transcripts included in the annotation²⁷.”

Reviewer #2:

Remarks to the Author:

The manuscript by Chung, Shen and colleagues presented the largest ASD gene discovery to date. It is a rigorous study with interesting findings, particularly about inherited LoF variants. Notably, it reported that while known ASD/DDD genes explained the majority of DNV burden, they explain only 20% of burden in inherited LOFs, suggesting many missing ASD genes. It also shows that it is possible to discover ASD genes using rare LoF variants from families or case-control studies, a finding that has implications on future study design. Overall, the interesting findings as well as the resources provided in this paper would make it appealing to the audience of Nature Genetics.

Major comments:

1) The main analysis is done in two stages, with the first stage identifying about 400 genes, and then in the second stage, meta-analysis is done on these genes to combine evidence from multiple tests. I don't see the benefit of this two-stage analysis, however. I would imagine that the smaller gene set from the first stage may reduce multiple testing burden in the second stage, potentially increasing the power of study. But in fact, in the second stage of meta-analysis, the authors still apply exome-wide significance threshold, so it doesn't really benefit from gene selection in the first stage. It seems straightforward to just apply the meta-analysis on all genes. This needs to be better explained.

Response: This is indeed an important point. The 404 genes were selected based on the results from Stage 1 among 5,754 constrained genes (ExAC pLI \geq 0.5 or top 20% of LOEUF scores). Because the data from Stage 1 were used in the overall analysis, there are two options to perform a multi-test adjustment. In the initial submission, we used the more conservative approach and applied an exome-wide significance threshold to nominate candidate genes. We

13opted for this approach in order to reduce any spurious results. Another option would be to use “study-wide significance”, accounting for all 5,754 autosomal constrained genes. In the revision, we inserted a sentence describing the 72 genes that reached study-wide significance ($p < 8.69 \times 10^{-6}$). We also added a column in **Supplementary Table S9** to indicate study-wide significance. We edited the following line on page 6 in the manuscript to describe the results based on study-wide significance: “We identified 60 genes with exome-wide significance ($p < 2.5 \times 10^{-6}$) and 72 genes reached study-wide significance accounting for all 5,754 constraint genes ($p < 8.69 \times 10^{-6}$, **Supplementary Table S9**).”

Unfortunately, it is not straightforward to conduct a meta-analysis on all genes. We will conduct a meta-analysis on all genes at a future stage but at the time of this analysis, the high-quality data required for each gene were not available.

2) The authors used a simple binomial test for LoF variants in case-control data. The concern here is that the cases and controls are from two different studies with potentially different biases and rates of calling LoF variants. This may lead to inflated p-values. The authors said that AFs of the LoF variants have similar distributions in cases and controls. This is certainly helpful, still, I think it would be more rigorous to show directly that the results are not inflated. A QQ plot of p-values from all genes would be helpful.

Response: Thank you for this comment. We are not able to produce a QQ plot of p-values from all genes and a QQ plot for the 404 genes would be inflated since we selected significant genes based on the results from Stage 1. Instead, we have generated a QQ plot comparing rare synonymous variants in the prioritized 404 genes between cases and controls (**Supplementary Figure S11**). Such a comparison can shed light on the degree of batch effects that may be due to differences in either technical properties of the data or population composition. Therefore, we compared the burden of ultra-rare synonymous variants (allele frequency $< 1 \times 10^{-5}$) between ASD cases and gnomAD controls in the case-control analysis between the 404 selected genes (**Supplementary Figure S11** below). The overall ultra-rare synonymous rate is 0.57 for both ASD cases and gnomAD controls for all ancestral groups. Most genes showed similar burdens between cases and controls. These results show that both LoF and synonymous variants have a similar distribution between cases and controls and support our conclusions. We added the following line on page 6 in the manuscript to describe the ultra-rare synonymous burden: “The ultra-rare synonymous variant burden is similar between cases and controls across the selected genes (**Supplementary Figure S11**).”

Supplementary Figure S11: QQ plot showing the ultra-rare synonymous burden test between SPARK cases and gnomAD controls. Panel A: cross-ancestry case-control ultra-rare synonymous burden comparison; Panel B: European-only case-control ultra-rare synonymous burden comparison.

3) In the meta-analysis, the three tests are combined using Fisher's method. While this is understandable, given that standard fixed effect meta-analysis is not applicable, Fisher's method is not optimal, especially when combining tests with different power. This may happen for example, for the genes that are driven by LoF DNVs, but contain very few inherited LoFs, thus the TDD/case-control test would have low power. It would be useful to assess how much is gained from meta-analysis. For example, if one just uses DNV data, how many genes are found? Similarly, how many are found using only inherited LoF variants (combining TDT and case-control)? How many genes are found using a single test, but actually missing in meta-analysis?

Response: We now included a supplementary table showing the results of DNVs, inherited variants and case-control analysis to clarify what genes are significant by only one analysis and by combined p-values (**Supplementary Table S9**). By using only DNV data among the selected 404 genes, 15 genes reached exome-wide significance (*DNMT3A, SCN2A, SLC6A1, PTEN, KDM5B, DEAF1, FOXP1, CHD8, KCNQ3, MYT1L, CHD2, TBL1X R1, SHANK3, PPP2R5D, GRIN2B*). By combining TDT and case-control analyses, 6 genes reached exome-wide significance (*SHANK3, NAV3, KDM5B, WDFY3, SPTBN1, DYNC1H1*). Importantly, there are no genes that are identified as significant in a single test but are not exome-wide significant in the meta-analysis.

154) The archetypal analysis is interesting. It nicely projects the functions of genes into 6 archetypes. However, the method is not explained clearly. This is especially a concern, given that this is not a common analysis (at least in the human genetics literature). Some specific comments and suggestions include: it would be useful to give a high-level description of the archetypal analysis to the readers (as a general background). In Table S10, there are 500 clusters. I suppose these clusters are found by STRING, but 500 seems a very large number of clusters. How are these clusters defined? What are the genes in each cluster? Also some rows of table S10 contain a single cluster, while others have more, why? How did the authors obtain "description" of each row?

Response:

We thank the reviewer for their interest in the archetypal analysis as well as their observations and suggestions to make this analysis more parsimonious and easier to follow. We have addressed these comments through the following changes: first, we have added to the methods a more thorough description of archetypal analysis as applied here:

Archetypal analysis¹⁰² (AA) is an unsupervised learning approach that has similarities to other dimensionality reduction and clustering approaches. An important distinction of AA from comparable approaches is that it seeks a set of k archetypes, which are points along the convex hull of the data, from which all data points may be expressed as a mixture. The output of AA is a N by k matrix, α , of $[0, 1]$ coefficients that represent the contribution of each archetype to each data point. While cluster centroids are embedded within the interior of a cluster, archetypes are, by design, at the extremes of the data (i.e., along the convex hull). In practice, this ensures that the archetypes are appreciably distinct from one another, which is often not the case of cluster centroids. Here, we use AA as a means to organize putative ASD risk genes in a space that has meaning from a mechanistic (STRING) and phenotypic (HPO) standpoint. Genes of particular interest to this study can then be considered and interpreted in the context of the archetypes that define this space

Second, we have revised and simplified the enrichment analyses, now using annotations from MSigDB v7.4, which were not used to define the archetypes. Only gene sets that were represented among the $N=62$ risk genes were investigated. This has the effect of focusing the interpretation on the most supportable risk pathways while also shortening the list of enriched annotations presented in the **Supplementary Table S13-S15**. While the focus has now shifted to the MSigDB analysis, we have retained the STRING cluster analysis results for the sake of

16completeness, and it is referenced briefly in the current text. To the reviewer's specific questions here, the clusters are defined by the STRING team as in [PMID: 33237311]. Briefly, they are localized regions of increased connectivity within the global STRING network, and were given human-readable descriptions by the STRING team, which we use in our table. Mappings of clusters to genes are also provided by STRING and are shown in our table. This table is now provided as **Supplementary Table S14**.

5) Another concern about the archetypal analysis: the authors have simple interpretations of each archetype, e.g. A6 is for KRAB domain and LRR. This makes it easy to follow the functions of genes. However, I feel that this may be an over-simplification. From Table S10: some archetypes seems to be quite functionally heterogeneous. For example, the authors annotate A4 as "membrane trafficking and protein transport", but the clusters associated with A4 have many other functions, particularly a number of signaling pathways such as Wnt, JAK-STAT, TGF-beta, interferon and BMP signaling. A6 seems to be even more heterogeneous, with functions including Olfactory signaling, homeodomain, Ca2 binding, cell polarity, glycosylation and so on. So annotating it as "KRAB domain and LRR" seems a bit misleading here.

Response: We agree that the archetypes show enrichment for multiple intersecting processes and mechanisms, and are grateful that the reviewer appreciates the challenge of making intuitive descriptors for these archetypes while also not over-simplifying the heterogeneity. In light of our revised analysis (i.e., focusing only on genes significantly linked to autism through rare variant association), we applied the following heuristic to guide the naming of each archetype: each MSigDB that has two or more autism genes (N=62 as described above) annotated is tested for association with the archetypal coefficients (A1-A6, simultaneously) in a quasibinomial GLM. The one-sided association p-value (i.e., positive association) for each archetype is reported in **Supplementary Table S15** (top 20 for each archetype). To name each archetype, the top 20 gene sets with the strongest associations with the other archetypes were considered: this leads to the following naming conventions: A1: neurotransmission; A2: chromatin modification; A3: RNA processing; A4: vesicle-mediated transport; A5: MAPK signaling and migration; A6: cytoskeleton and mitosis (see **Supplementary Table S12**). In addition, we have updated the manuscript text to make clear to the reader that the names of the archetypes are not meant to imply perfect functional homogeneity among genes affiliated with those archetypes, rather, they represent the most general and supportable mechanistic theme that is represented among the genes with elevated scores for the corresponding archetype. We feel this heuristic arrived at mostly interpretable and generalizable names that do justice to the underlying mechanistic heterogeneity while offering the reader something to latch onto beyond the numbers or colors often used in naming modules in similar analyses, but that we find to be too abstracted from the interpretation.

Minor comments

17Fig. 3 and 4 are in the wrong order.

Response: This is now corrected.

Fig. 4: right panel, "percentage of over-transmission of rare LoFs to cases". Why the percents there are sometimes so high, reaching 50-60%? How is "over-transmission" defined? If I understand correctly, it should be percent of transmitted vs. non-transmitted, but then 50-60% seems too high. Fig. 2B, for examples, show much smaller over-transmission.

Response: In **Figure 2B**, we show the estimated number of over-transmission per trio, while in **Figure 3B** (Figure 3 was previously labeled Figure 4) shows the fraction of over-transmission events in a selected gene set over all over-transmission events among all constraint genes. We clarified the x-axis label as "Share of Over-transmission" in **Figure 3B** and edited the legends as below:

"Enrichment of rare LoF variants in ASD cases across gene sets. Gene sets were defined and grouped by transcriptome proteome, neuronal regulome, ASD gene prediction scores, genetic evidence from neuropsychiatric diseases, and gene level constraint. Analyses were repeated after removing known ASD/NDD genes. (Number of genes in each set before and after removing known genes are shown in bracket below gene set.) Dots represent fold enrichment of DNVs or odds ratios for over-transmission of LoFs in each set. Horizontal bars indicate the 95% confidence interval. For each gene set, we show the percentage of over-transmission of rare LoFs to cases. Enrichment of rare, inherited LoFs was evaluated by the share of over-transmission events (the transmission and non-transmission of ultra-rare LoFs with $p_{Ext} \geq 0.1$) in the selected gene set versus those in all other constrained genes using a 2-by-2 table. P-values were given using the chi-squared test."

Fig. 5 legend: in my Word file, I see, "All five novel genes (Error! Reference source not found)".

Response: This is now corrected.

Decision Letter, first revision:

Our ref: NG-A58552R

25th Feb 2022

Dear Dr. Feliciano,

18Thank you for submitting your revised manuscript "Integrating de novo and inherited variants in over 42,607 autism cases identifies mutations in new moderate risk genes" (NG-A58552R). It has now been seen by the original referees and their comments are below. The reviewers find that the paper has improved in revision, and therefore we'll be happy in principle to publish it in Nature Genetics, pending minor revisions to satisfy the referees' final (minor) requests (Reviewer #2 only) and to comply with our editorial and formatting guidelines.

Sincerely,

Safia Danovi
Editor
Nature Genetics

Reviewer #1 (Remarks to the Author):

The authors have addressed most of my comments satisfactorily.

Reviewer #2 (Remarks to the Author):

The authors have addressed most of my comments. I have only one minor comment left:

Line 1248-1249: "To name each archetype, the top 20 gene sets with the strongest associations with the other archetypes were considered". This sentence reads strange, why "associations with the other archetypes"? I suppose to name each archetype, one finds the top 20 gene sets associated with that archetype.

Final Decision Letter:

19In reply please quote: NG-A58552R1 Chung

28th Jun 2022

Dear Dr. Chung,

I am delighted to say that your manuscript "Integrating de novo and inherited variants in over 42,607 autism cases identifies mutations in new moderate risk genes" has been accepted for publication in an upcoming issue of Nature Genetics.

Your paper will be published online after we receive your corrections and will appear in print in the next available issue. You can find out your date of online publication by contacting the Nature Press Office (press@nature.com) after sending your e-proof corrections. Now is the time to inform your Public Relations or Press Office about your paper, as they might be interested in promoting its publication. This will allow them time to prepare an accurate and satisfactory press release. Include your manuscript tracking number (NG-A58552R1) and the name of the journal, which they will need when they contact our Press Office.

20Please note that *Nature Genetics* is a Transformative Journal (TJ). Authors may publish their research with us through the traditional subscription access route or make their paper immediately open access through payment of an article-processing charge (APC). Authors will not be required to make a final decision about access to their article until it has been accepted. [Find out more about Transformative Journals](https://www.springernature.com/gp/open-research/transformative-journals)

Authors may need to take specific actions to achieve [compliance with funder and institutional open access mandates](https://www.springernature.com/gp/open-research/funding/policy-compliance-faqs). If your research is supported by a funder that requires immediate open access (e.g. according to [Plan S principles](https://www.springernature.com/gp/open-research/plan-s-compliance)) then you should select the gold OA route, and we will direct you to the compliant route where possible. For authors selecting the subscription publication route, the journal's standard licensing terms will need to be accepted, including [self-archiving and license to publish](https://www.nature.com/nature-portfolio/editorial-policies/self-archiving-and-license-to-publish). Those licensing terms will supersede any other terms that the author or any third party may assert apply to any version of the manuscript.

Please note that Nature Portfolio offers an immediate open access option only for papers that were first submitted after 1 January, 2021.

If you have not already done so, we invite you to upload the step-by-step protocols used in this manuscript to the Protocols Exchange, part of our on-line web resource, natureprotocols.com. If you complete the upload by the time you receive your manuscript proofs, we can insert links in your article that lead directly to the protocol details. Your protocol will be made freely available upon publication of your paper. By participating in natureprotocols.com, you are enabling researchers to more readily reproduce or adapt the methodology you use. [Natureprotocols.com](http://natureprotocols.com) is fully searchable, providing your protocols and paper with increased utility and visibility. Please submit your protocol to <https://protocolexchange.researchsquare.com/>. After entering your nature.com username and password you will need to enter your manuscript number (NG-A58552R1). Further information can be found at <https://www.nature.com/nature-portfolio/editorial-policies/reporting-standards#protocols>

Sincerely,

Safia Danovi
Editor
Nature Genetics